# CoCoNuTs are a diverse subclass of Type IV restriction systems predicted to target RNA

Ryan T Bell*, Harutyun Sahakyan, Kira S Makarova, Yuri I Wolf, Eugene V Koonin*

National Center for Biotechnology Information, National Library of Medicine, National Institutes of Health, Bethesda, United States

**\*For correspondence:**
ryan.bell@nih.gov (RTB);
koonin@ncbi.nlm.nih.gov (EVK)

**Competing interest:** The authors declare that no competing interests exist.

**Abstract** A comprehensive census of McrBC systems, among the most common forms of prokaryotic Type IV restriction systems, followed by phylogenetic analysis, reveals their enormous abundance in diverse prokaryotes and a plethora of genomic associations. We focus on a previously uncharacterized branch, which we denote *coiled-coil nuclease tandems* (CoCoNuTs) for their salient features: the presence of extensive coiled-coil structures and tandem nucleases. The CoCoNuTs alone show extraordinary variety, with three distinct types and multiple subtypes. All CoCoNuTs contain domains predicted to interact with translation system components, such as OB-folds resembling the SmpB protein that binds bacterial transfer-messenger RNA (tmRNA), YTH-like domains that might recognize methylated tmRNA, tRNA, or rRNA, and RNA-binding Hsp70 chaperone homologs, along with RNases, such as HEPN domains, all suggesting that the CoCoNuTs target RNA. Many CoCoNuTs might additionally target DNA, via McrC nuclease homologs. Additional restriction systems, such as Type I RM, BREX, and Druantia Type III, are frequently encoded in the same predicted superoperons. In many of these superoperons, CoCoNuTs are likely regulated by cyclic nucleotides, possibly, RNA fragments with cyclic termini, that bind associated CARF (*C*RISPR-*A*ssociated *R*ossmann *F*old) domains. We hypothesize that the CoCoNuTs, together with the ancillary restriction factors, employ an echeloned defense strategy analogous to that of Type III CRISPR-Cas systems, in which an immune response eliminating virus DNA and/or RNA is launched first, but then, if it fails, an abortive infection response leading to PCD/dormancy via host RNA cleavage takes over.

## eLife assessment

This article marks a **fundamental** advance in our understanding of prokaryotic Type IV restriction systems. The authors provide an encyclopedic overview of a hitherto uncharacterized branch of these systems, which they name CoCoNuTs, for coiled-coil nuclease tandems. They provide **compelling** evidence that these nucleases target RNA and are part of an echeloned defense response following viral infection. This article will be of great interest to scientists studying prokaryotic immunity mechanisms, as well as broadly to protein scientists engaged in the analysis, classification, and functional annotation of the proteome of life.

## Introduction

All organisms are subject to an incessant barrage of genetic parasites, such as viruses and transposons. Over billions of years, the continuous arms race between hosts and parasites drove the evolution of immense, intricately interconnected networks of diverse defense systems and pathways (*Burroughs et al., 2015*; *Gao et al., 2020*; *Goldfarb et al., 2015*; *Bell et al., 2020*; *Koonin and Aravind, 2002*;

**eLife digest** All organisms, from animals to bacteria, are subject to genetic parasites, such as viruses and transposons. Genetic parasites are pieces of nucleic acids (DNA or RNA) that can use a cell's machinery to copy themselves at the expense of their hosts. This often leads to the host's demise, so organisms evolved many types of defense mechanisms. One of the most ancient and common forms of defense against viruses and transposons is the targeted restriction of nucleic acids, that is, deployment of host enzymes that can destroy or restrict nucleic acids containing specific sequence motifs or modifications.

In bacteria, many of the restriction enzymes targeting parasitic genetic elements are formed by fusions of proteins from the so-called McrBC systems with a protein domain called EVE. EVE and other functionally similar domains are a part of proteins that recognize and bind modified bases in nucleic acids. Enzymes can use the ability of these specificity domains to bind modified bases to detect non-host nucleic acids.

Bell et al. conducted a comprehensive computational search for McrBC systems and discovered a large and highly diverse branch of this family with unusual characteristic structural and functional domains. These features include regions that form long alpha-helices (coils) that coil with other alpha-helices (known as coiled-coils), as well as several distinct enzymatic domains that break down nucleic acids (known as nucleases). They call these systems CoCoNuTs (*c*oiled-*c*oiled *nu*clease *t*andems).

All CoCoNuTs contain domains, including EVE-like ones, which are predicted to interact with components of the RNA-based systems responsible for producing proteins in the cell (translation), suggesting that the CoCoNuTs have an important impact on protein abundance and RNA metabolism.

Bell et al.'s findings will be of interest to scientists working on prokaryotic immunity and virulence. Furthermore, similarities between CoCoNuTs and components of eukaryotic RNA-degrading systems suggest evolutionary connections between this diverse family of bacterial predicted RNA restriction systems and RNA regulatory pathways of eukaryotes.

Further deciphering the mechanisms of CoCoNuTs could shed light on how certain pathways of RNA metabolism and regulation evolved, and how they may contribute to advances in biotechnology.

*Swarts et al., 2014*). In particular, in the last few years, targeted searches for defense systems in prokaryotes, typically capitalizing on the presence of variable genomic defense islands, have dramatically expanded their known diversity and led to the discovery of a plethora of biological conflict strategies and mechanisms (*Gao et al., 2020*; *Bell et al., 2020*; *Anantharaman et al., 2012*; *Kaur et al., 2020*).

One of the most ancient and common forms of defense against mobile genetic elements (MGE) is the targeted restriction of nucleic acids. Since the initial discovery of this activity among strains of bacteria resistant to certain viruses, myriad forms of recognition and degradation of nucleic acids have been described, in virtually all life forms. Characterization of the most prominent of these systems, such as restriction-modification (RM), RNA interference (RNAi), and CRISPR-Cas (*c*lustered *r*egularly *i*nterspaced *s*hort *p*alindromic *r*epeats-*C*RISPR *a*ssociated genes), has led to the development of a profusion of highly effective experimental and therapeutic techniques, in particular, genome editing and engineering (*Loenen et al., 2014*; *Fire et al., 1998*; *Agrawal et al., 2003*; *Makarova et al., 2006*; *Makarova et al., 2020b*; *Gasiunas et al., 2012*; *Jinek et al., 2012*).

Historically, the two-component McrBC (*m*odified *c*ytosine *r*estriction) system was the first form of restriction to be described, although the mechanism remained obscure for decades, and for a time, this system was referred to as RglB (*r*estriction of *g*lucose*l*ess phages) due to its ability to restrict T-even phage DNA which contained hydroxymethylcytosine, but not glucosylated bases (*Raleigh et al., 1989*; *Luria and Human, 1952*; *Fleischman et al., 1976*; *Dila et al., 1990*). Today, the prototypical McrBC system, native to *Escherichia coil* K-12, is considered a Type IV (modification-dependent) restriction system that degrades DNA containing methylcytosine (5mC) or hydroxymethylcytosine (5hmC), with a degree of sequence context specificity (*Sutherland et al., 1992*; *Sukackaite et al., 2012*).

Type IV restriction enzymes contain at least two components: (1) a dedicated specificity domain that recognizes modified DNA and (2) an endonuclease domain that cleaves the target (*Weigele and Raleigh, 2016*; *Loenen et al., 2014*). In the well-characterized example from *E. coli* K-12, McrB harbors

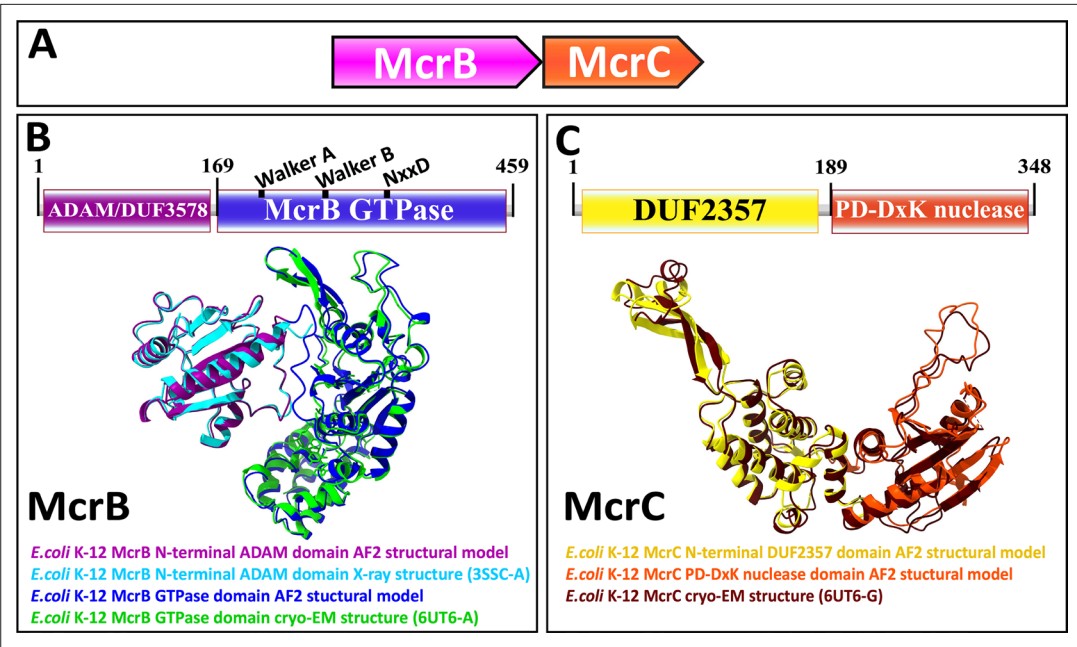

**Figure 1.** Genetic organization, signature sequence motifs, structural models, and phyletic distribution of McrB GTPases detected in this work. (**A**) McrBC is a two-component restriction system with each component typically (except for extremely rare gene fusions) encoded by a separate gene expressed as a single operon, depicted here and in subsequent figures as arrows pointing in the direction of transcription. In most cases, McrB is the upstream gene in the operon. (**B**) In the prototypical *E. coli* K-12 McrBC system, McrB contains an N-terminal methylcytosine-binding domain, ADAM/DUF3578, fused to a GTPase of the AAA+ATPase clade (*Sukackaite et al., 2012*). This GTPase contains the Walker A and Walker B motifs that are conserved in P-loop NTPases as well as a signature NxxD motif, all of which are required for GTP hydrolysis (*Nirwan et al., 2019*; *Niu et al., 2020*; *Pieper et al., 1999*). An AlphaFold2 structural model of *E. coli* K-12 ADAM-McrB GTPase fusion protein monomer and separate X-ray diffraction and cryo-EM structures of the ADAM and GTPase domains (*Niu et al., 2020*; *Sukackaite et al., 2012*) show a high degree of similarity. (**C**) McrC consists of a PD-DxK nuclease and an N-terminal DUF2357 domain, which comprises a helical bundle with a stalk-like extension that interacts with and activates individual McrB GTPases while they are assembled into hexamers (*Niu et al., 2020*; *Nirwan et al., 2019*). An AlphaFold2 structural model and cryo-EM structure of *E. coli* K-12 McrC monomer with DUF2357-PD-DxK architecture (*Niu et al., 2020*) show a high degree of similarity. The structures were visualized with ChimeraX (*Pettersen et al., 2021*).

an N-terminal DUF (domain of unknown function) 3578 that recognizes methylcytosine. We denote this domain, as its function is not unknown, as ADAM (*a domain with an affinity for methylcytosine*). The ADAM domain is fused to a GTPase domain of the AAA+ATPase superfamily, the only known GTPase in this clade, which is believed to translocate DNA (*Figure 1A and B*; *Sutherland et al., 1992*; *Iyer et al., 2004a*; *Nirwan et al., 2019*; *Panne et al., 1999*). McrC, encoded by a separate gene in the same operon, contains an N-terminal DUF2357 domain, which interacts with the GTPase domain in McrB and stimulates its activity, and is fused to a PD-(D/E)xK superfamily endonuclease (*Figure 1A and C*; *Niu et al., 2020*; *Sutherland et al., 1992*).

Our recent analysis of the modified base-binding EVE (named for Protein Data Bank PDB structural identifier 2eve) domain superfamily demonstrated how the distribution of the EVE-like domains connects the elaborate eukaryotic RNA regulation and RNA interference-related epigenetic silencing pathways to largely uncharacterized prokaryotic antiphage restriction systems (*Bell et al., 2020*). EVE superfamily domains, which in eukaryotes recognize modified DNA or RNA as part of mRNA maturation or epigenetic silencing functions, are often fused to McrB-like GTPases in prokaryotes, and indeed, these are the most frequently occurring EVE-containing fusion proteins (*Bell et al., 2020*). These observations motivated us to conduct a comprehensive computational search for McrBC systems, followed by a census of all associated domains, to chart the vast and diverse population of antiviral specificity modules, vital for prokaryotic defense, that also provided important source material

during the evolution of central signature features of eukaryotic cells. Here we present the results of this census and describe an extraordinary, not previously appreciated variety of domain architectures of the McrBC family of Type IV restriction systems. In particular, we focus on a major McrBC branch that we denote *coiled-coil nuclease tandem* (CoCoNuT) systems, which we explore in detail.

## Results and discussion
### Comprehensive census of McrBC systems

The search for McrBC systems included PSI-BLAST runs against the non-redundant protein sequence database at the NCBI, followed by several filtering strategies (see 'Methods') to obtain a clean set of nearly 34,000, distributed broadly among prokaryotes. In the subsequent phase of analysis, GTPase domain sequences were extracted from the McrB homolog pool, and DUF2357 and PD-(D/E)xK nuclease domain sequences were extracted from the McrC homolog pool, leaving as a remainder the fused specificity domains that we intended to classify (although the McrC homologs are not the primary bearers of specificity modules in McrBC systems, they can be fused to various additional domains, including those of the EVE superfamily) (*Figure 2*, *Figure 2—figure supplement 1*). Unexpectedly, we

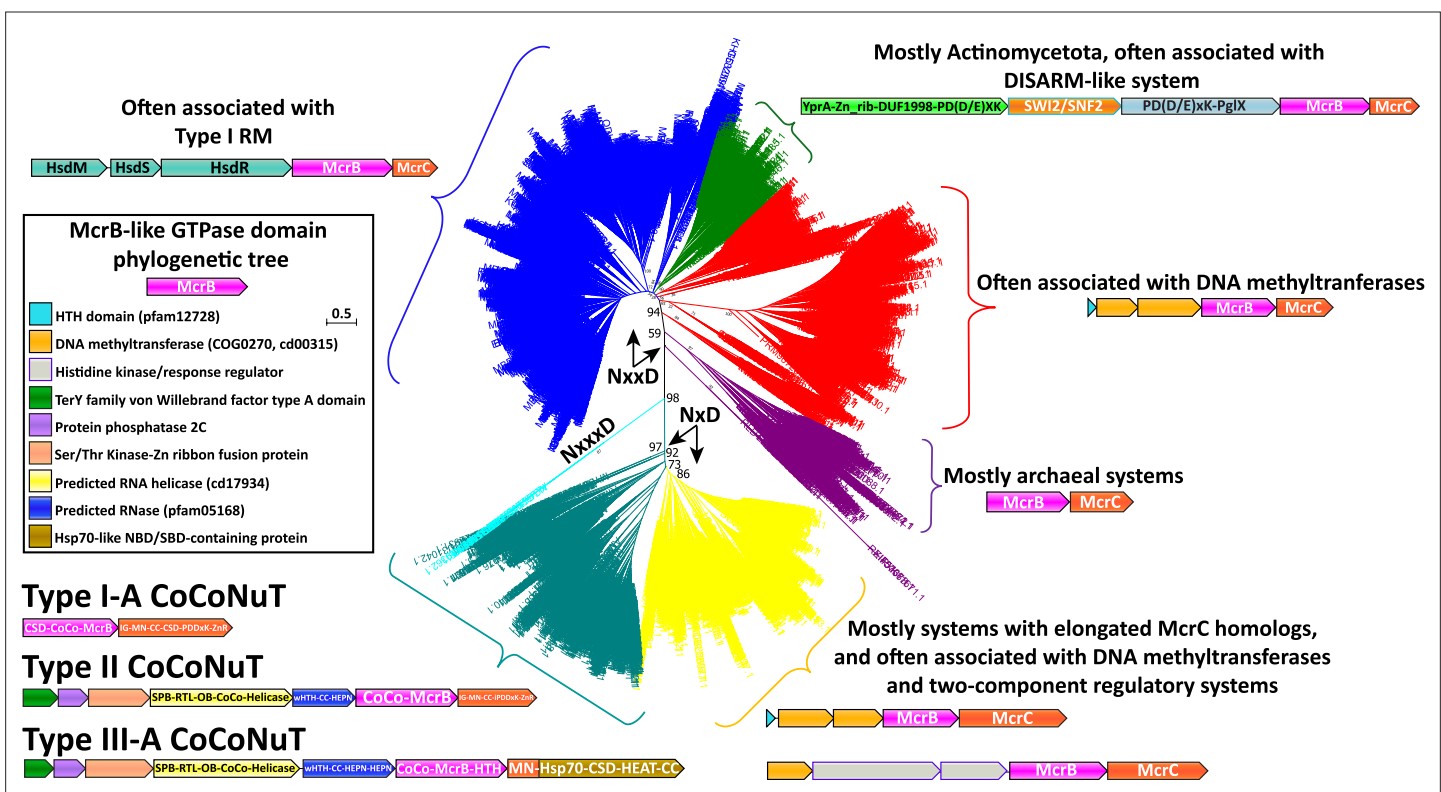

**Figure 2.** Phylogenetic tree of the McrB-like GTPases. The major clades in the phylogenetic tree of the McrB-like GTPases are distinguished by the distinct versions of the Nx(xx)D signature motif. The teal and yellow groups, with bootstrap support of 97%, have an NxD motif, whereas the blue, green, red, and purple groups, with variable bootstrap support, have an NxxD motif, indicated by the arrows; the sequences in the smaller, cyan clade, with 98% bootstrap support, have an NxxxD motif. Each of the differently colored groups is characterized by distinct conserved genomic associations that are abundant within but not completely confined to the respective groups. This tree was built from the representatives of 90% identity clusters of all validated homologs. Abbreviations of domains: McrB, McrB-like GTPase domain; CoCo/CC, coiled-coil; MN, McrC N-terminal domain (DUF2357); CSD, cold shock domain; IG, immunoglobulin (IG)-like beta-sandwich domain; ZnR, zinc ribbon domain; SPB, SmpB-like domain; RTL, RNase toxin-like domain; HEPN, HEPN family nuclease domain; OB, OB-fold domain; iPD-(D/E)xK, inactivated PD-(D/E)xK fold; Hsp70, Hsp70-like NBD/SBD; HEAT, HEAT-like helical repeats; YprA, YprA-like helicase domain; DUF1998, DUF1998 is often found in or associated with helicases and contains four conserved, putatively metal ion-binding cysteine residues; SWI2/SNF2, SWI2/SNF2-family ATPase; PglX, PglX-like DNA methyltransferase; HsdR/M/S, Type I RM system restriction, methylation, and specificity factors.

The online version of this article includes the following figure supplement(s) for figure 2:

**Figure supplement 1.** Phylogenetic tree of the McrC DUF2357 domain.

found that both the GTPase domains and DUF2357 domains frequently contained insertions into their coding sequences, likely encoding specificity domains and coiled-coils, respectively.

Removing variable inserts from the conserved McrBC domains allowed accurate, comprehensive phylogenetic analysis of the McrB GTPase (*Figure 2*) and McrC DUF2357 (*Figure 2—figure supplement 1*) families. These two trees were generally topologically concordant and revealed several distinct branches not previously recognized. The branch containing the prototypical McrBC system from *E. coil* K-12 (*Figure 2*, blue) is characterized by frequent genomic association with Type I RM systems, usually with unidirectional gene orientations and the potential of forming a single operon (*Raleigh, 1992*). This branch and others, which exhibit two particularly prevalent associations, with a DISARM-like antiphage system (*Figure 2*, green; *Ofir et al., 2018*), and, surprisingly for a Type IV restriction system, with predicted DNA methyltransferases (*Figure 2*, red), will be the subject of a separate, forthcoming publication.

## A variant of the McrB GTPase signature motif distinguishes a large group of unusual McrBC systems

A major branch (*Figure 2*, teal and yellow) of the McrBC family is characterized by a conserved deletion within the NxxD GTPase signature motif (where x is any amino acid), found in all McrB-like GTPases, reducing it to NxD (*Figures 1B and 2*; *Neuwald et al., 1999*; *Iyer et al., 2004a*; *Erzberger and Berger, 2006*; *Niu et al., 2020*). The NxD variant of the motif is strictly conserved in these homologs and is usually, but not invariably, followed by a glutamate (E) or a second aspartate (D). No NxD motif McrB GTPase has been characterized, but the extensive study of the NxxD motif offers clues to the potential impact of the motif shortening. The asparagine (N) residue is strictly required for GTP binding and hydrolysis (*Pieper et al., 1999*). As this residue is analogous to sensor-1 in ATP-hydrolyzing members of the AAA+family, it can be predicted to position a catalytic water molecule for nucleophilic attack on the γ-phosphate of an NTP (*Erzberger and Berger, 2006*; *Colicelli, 2004*; *Bourne et al., 1991*; *Niu et al., 2020*; *Pieper et al., 1999*). A recent structural analysis has shown that the aspartate interacts with a conserved arginine/lysine residue in McrC which, via a hydrogen-bonding network, resituates the NxxD motif in relation to its interface with the Walker B motif such that, together, they optimally position a catalytic water to stimulate hydrolysis (*Niu et al., 2020*). Accordingly, the truncation of this motif might be expected to modulate the rate of hydrolysis and potentially compel functional association with only a subset of McrC homologs containing compensatory mutations. The NxD branch is characterized by many McrC homologs with unusual features, such as predicted RNA-binding domains, that might not be compatible with conventional McrB GTPases from the NxxD clade, which often occur in the same genomes (see below).

Many of these NxD GTPases are contextually associated with DNA methyltransferases, like the NxxD GTPases, but are distinguished by additional complexity in the domains fused to the McrC homologs and the frequent presence of two-component regulatory system genes in the same operon (*Figure 2*, yellow). We also detected a small number of GTPases with an insertion in the signature motif (*Figure 2*, cyan), expanding it to NxxxD, which are methyltransferase-associated as well. This association is likely to be ancestral as it is found in all three branches of the McrB GTPase tree with different signature motif variations.

The most notable feature of the NxD branch of GTPases is a large clade characterized by fusion to long coiled-coil domains (*Figure 2*, teal; *Supplementary file 1*), a derived and distinctly different architecture from the small, modular DNA-binding specificity domains typically found in canonical McrB homologs. The McrC homologs associated with these coiled-coil McrB-like GTPase fusions also often lack the PD-(D/E)xK endonuclease entirely or contain a region AlphaFold2 (AF2) predicted to adopt the PD-(D/E)xK endonuclease fold, but with inactivating replacements of catalytic residues (*Supplementary file 1*; *Jumper et al., 2021*). However, in the cases where the McrC nuclease is missing or likely inactive, these systems are always encoded in close association, often with overlapping reading frames, with HEPN (*h*igher *e*ukaryotes and *p*rokaryotes *n*ucleotide-binding) ribonuclease domains, or other predicted nucleases (*Figure 3*, *Figure 3—figure supplement 1*; *Pillon et al., 2021*). Based on these features of domain architecture and genomic context, we denote these uncharacterized McrBC-containing operons CoCoNuT systems.

The deepest branching group of the NxD GTPase systems is typified by a fusion of an Hsp70-like ATPase nucleotide-binding domain and substrate-binding domain (NBD/SBD) to the McrB GTPase

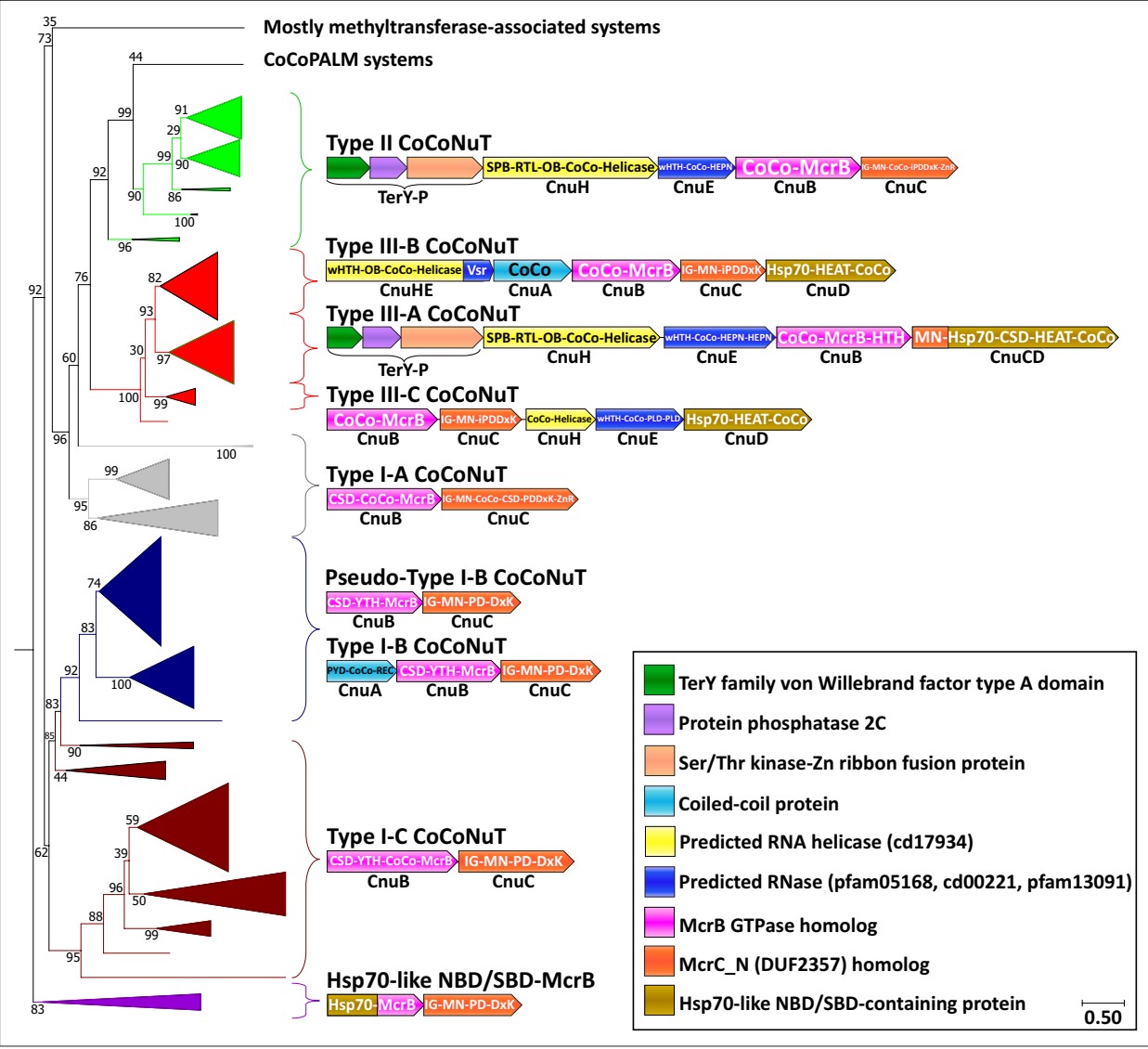

**Figure 3.** *C*oiled-*co*il *nu*clease *t*andem (CoCoNuT) system phylogeny and classification. The figure shows the detailed phylogeny of McrB-like GTPases from CoCoNuT systems and their close relatives. All these GTPases possess an NxD GTPase motif rather than NxxD. This tree was built from the representatives of 90% clustering of all validated homologs. Abbreviations of domains: McrB, McrB-like GTPase domain; CoCo, coiled-coil; MN, McrC N-terminal domain (DUF2357); CSD, cold shock domain; YTH, YTH-like domain; IG, immunoglobulin (IG)-like beta-sandwich domain; Hsp70, Hsp70-like NBD/SBD; HEAT, HEAT-like helical repeats; ZnR, zinc ribbon domain; PYD, pyrin/CARD-like domain; SPB, SmpB-like domain; RTL, RNase toxin-like domain; HEPN, HEPN family nuclease domain; OB, OB-fold domain; iPD-(D/E)xK, inactivated PD-(D/E)xK fold; REC, phosphoacceptor receiver-like domain; PLD, phospholipase D-like nuclease domain; Vsr, very-short-patch-repair PD-(D/E)xK nuclease-like domain. Underneath each gene is a proposed protein name, with Cnu as an abbreviation for *C*o*Co*N*u*T.

The online version of this article includes the following figure supplement(s) for figure 3:

**Figure supplement 1.** Phylogenetic tree of McrB family GTPases containing the NxD motif.

**Figure supplement 2.** Pseudo-Type I-B *c*oiled-*co*il *nu*clease *t*andem (CoCoNuT) genomic context in *Bacillus*.

domain (*Figure 3*, purple, *Figure 3—figure supplement 1*, purple, *Supplementary file 1*). Hsp70 is an ATP-dependent protein chaperone that binds exposed hydrophobic peptides and facilitates protein folding (*Mayer, 2021*). It also associates with AU-rich mRNA, and in some cases, such as the bacterial homolog DnaK, C-rich RNA, an interaction involving both the NBD and SBD (*Kishor et al., 2017*; *Zimmer et al., 2001*; *Kishor et al., 2013*). These systems might be functionally related to the CoCoNuTs, given that an analogous unit is encoded by a type of CoCoNuT system where similar domains are fused to the McrC homolog rather than to the McrB GTPase homolog (see below).

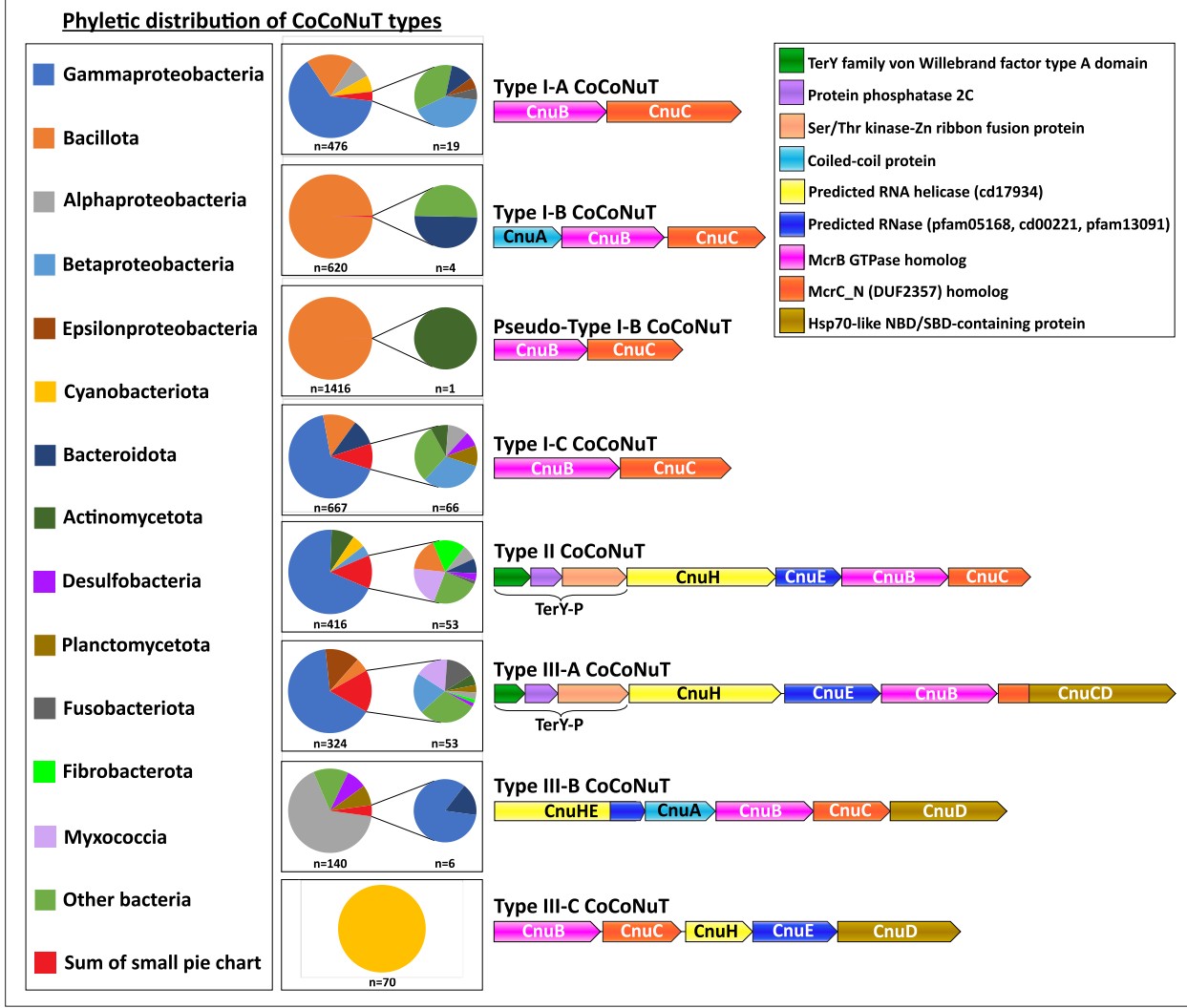

**Figure 4.** Phyletic distribution of *coiled-coil nuclease tandems* (CoCoNuTs). The phyletic distribution of CnuB/McrB-like GTPases in CoCoNuT systems found in genomic islands with distinct domain compositions. Most CoCoNuTs are found in either Bacillota or Pseudomonadota, with particular abundance in Gammaproteobacteria. Type I-B and the related Pseudo-Type I-B CoCoNuTs are restricted mainly to Bacillota. In contrast, the other types are more common in Pseudomonadota, but can be found in a wide variety of bacteria. Types III-B and III-C are primarily found in Alphaproteobacteria and Cyanobacteriota, respectively.

In another large clade of CoCoNuT-like systems, the GTPase is fused to a domain homologous to FtsB, an essential bacterial cell division protein containing transmembrane and coiled-coil helices (***Figure 3—figure supplement 1***, ***Supplementary file 1***; ***Khadria and Senes, 2013***). They are associated with signal peptidase family proteins likely to function as pilus assembly factors (***Supplementary file 1***; ***Colicelli, 2004***) and usually contain coiled-coil domains fused to both the McrB and McrC homologs. Therefore, we denote them *coiled-coil and pilus assembly linked to McrBC* (CoCoPALM) systems (***Figure 3—figure supplement 1***).

Here, we focus on the CoCoNuTs, whereas the CoCoPALMs and the rest of the NxD GTPase methyltransferase-associated homologs will be explored in a separate, forthcoming publication. CoCoNuT systems are extremely diverse and represented in a wide variety of bacteria, particularly Pseudomonadota and Bacillota, but are nearly absent in archaea (***Figure 4***).

## Type I CoCoNuT systems

We classified the CoCoNuTs into three types and seven subtypes based on the GTPase domain phylogeny and conserved genomic context (***Figure 3***). Type I-A and Type I-C systems consist of McrB

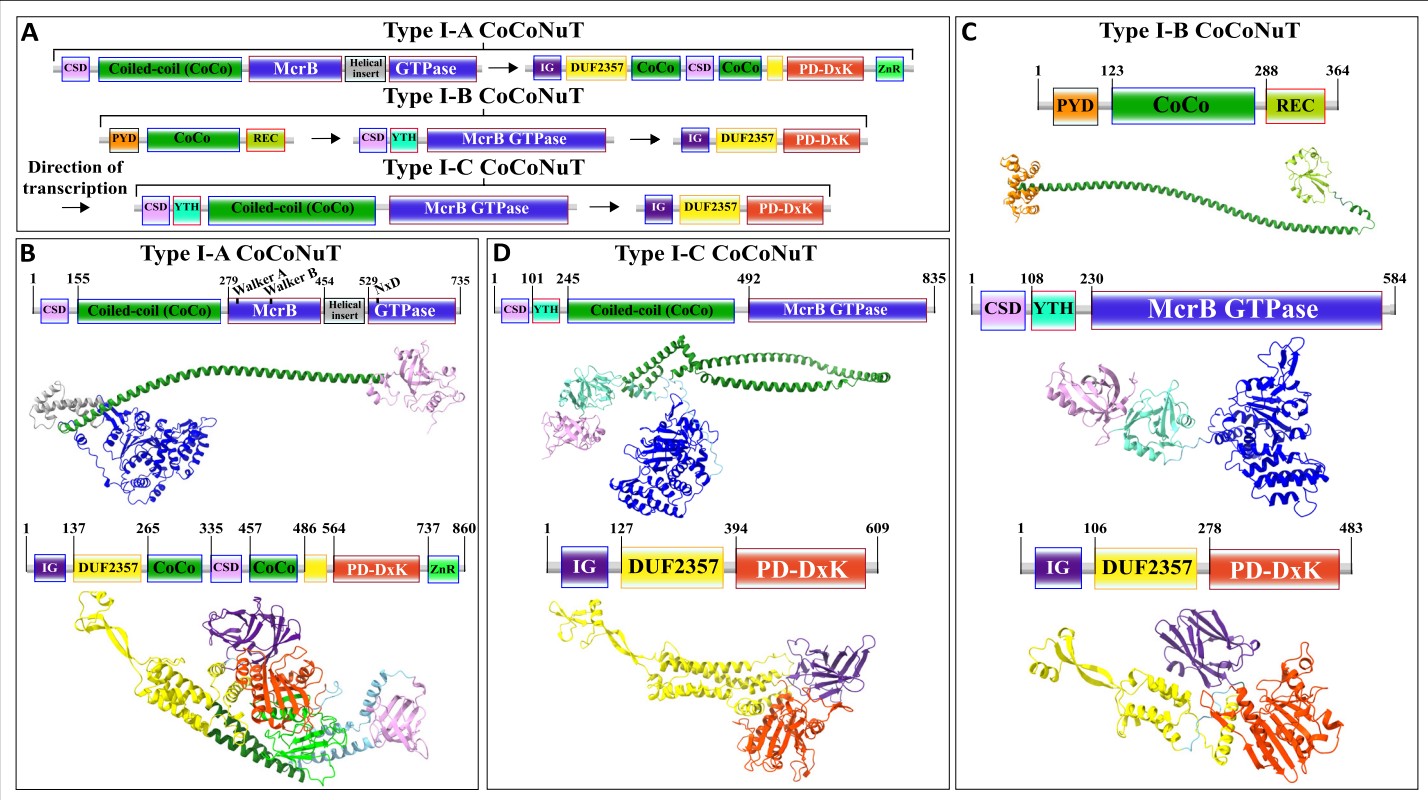

**Figure 5.** Domain composition, operon organization, and AlphaFold2 structural predictions of components of the Type I *coiled-coil nuclease tandem* (CoCoNuT) systems. (**A**) Type I CoCoNuT domain composition and operon organization. The arrows indicate the direction of transcription. (**B–D**) High-quality (average predicted local distance difference test [pLDDT] > 80), representative AlphaFold2 structural predictions for protein monomers in (**B**) Type I-A CoCoNuT systems (CnuB and CnuC, from top to bottom), (**C**) Type I-B CoCoNuT systems (CnuA, CnuB, and CnuC, from top to bottom), and (**D**) Type I-C CoCoNuT systems (CnuB and CnuC, from top to bottom). Models were generated from representative sequences with the following GenBank accessions (see ***Supplementary file 3*** for sequences and locus tags): ROR86958.1 (Type I-A CoCoNuT CnuB), APL73566.1 (Type I-A CoCoNuT CnuC), TKH01449.1 (Type I-B CoCoNuT CnuA), GED20858.1 (Type I-B CoCoNuT CnuB), GED20857.1 (Type I-B CoCoNuT CnuC), GFD85286.1 (Type I-C CoCoNuT CnuB), and MBV0932851.1 (Type I-C CoCoNuT CnuC). Abbreviations of domains: CSD, cold shock domain; YTH, YTH-like domain; CoCo, coiled-coil; IG, immunoglobulin (IG)-like beta-sandwich domain; ZnR, zinc ribbon domain; PYD, pyrin/CARD-like domain; REC, phosphoacceptor receiver-like domain. These structures were visualized with ChimeraX (***Pettersen et al., 2021***).

The online version of this article includes the following figure supplement(s) for figure 5:

**Figure supplement 1.** Domain composition and AlphaFold2 structural predictions of components of the Type I *coiled-coil nuclease tandem* (CoCoNuT) systems colored by pLDDT.

**Figure supplement 2.** AlphaFold2 prediction of Type I-A *coiled-coil nuclease tandem* (CoCoNuT) CnuB GTPase hexamer and CnuC monomer complex.

**Figure supplement 3.** AlphaFold2 prediction of Type I-B *coiled-coil nuclease tandem* (CoCoNuT) CnuB hexamer and CnuC monomer complex.

**Figure supplement 4.** AlphaFold2 prediction of Type I-C *coiled-coil nuclease tandem* (CoCoNuT) CnuB GTPase hexamer and CnuC monomer complex.

and McrC homologs only, which we denote CnuB and CnuC (***Figure 3*** and ***Figure 5***, ***Figure 5—figure supplements 1 and 2***, ***Figure 5—figure supplement 4***). Type I-A is distinguished from all other CoCoNuTs by a helical insert into the CnuB GTPase domain, between the Walker B and NxD motifs (***Figure 5***, ***Figure 5—figure supplement 2***).

Type I-B systems usually encode a separate coiled-coil protein, which we denote CnuA, in addition to CnuB/McrB and CnuC/McrC, with no coiled-coil fused to the GTPase domain in CnuB (***Figures 3 and 5***, ***Figure 5—figure supplement 3***). CnuA is fused at the N-terminus to a pyrin (PYD)/CARD (*caspase activation and recruitment domain*)-like helical domain and at the C-terminus to a phosphoacceptor receiver (REC) domain (***Figure 5***, ***Figure 5—figure supplement 1***, ***Supplementary file 1***). The association of the PYD/CARD-like domains with CoCoNuTs suggests involvement in a

programmed cell death (PCD)/abortive infection-type response as they belong to the DEATH domain superfamily and are best characterized in the context of innate immunity, inflammasome formation, and PCD (*Park et al., 2007*). The REC domains constitute one of the components of two-component regulatory systems. They are targeted for phosphorylation by histidine kinases (*Stock et al., 2000*), which could be a mechanism of Type I-B CoCoNuT regulation.

In contrast to Type II and Type III CnuB GTPase homologs, which contain only coiled-coils fused at their N-termini (*Figure 6*, *Figure 6—figure supplement 1*, *Supplementary file 1*), cold shock domain (CSD)-like OB-folds are usually fused at the N-termini of Type I CoCoNuT CnuB/McrB proteins, in addition to the coiled-coils (except for Type I-B, where the coiled-coils are encoded separately) (*Figure 5*, *Figure 5—figure supplement 1*, *Supplementary file 1*; *Amir et al., 2018*). Often, in Type I-B and I-C, but not Type I-A, a YTH-like domain, a member of the modified base-binding EVE superfamily, is present in CnuB as well, between the CSD and coiled-coil, or in Type I-B, between the CSD and the GTPase domain (*Figure 5*, *Figure 5—figure supplement 1*, *Supplementary file 1*; *Bell et al., 2020*; *Hazra et al., 2019*; *Liao et al., 2018*).

The CnuC/McrC proteins in Type I CoCoNuTs, as well as those in Type II and Types III-B and III-C, all contain an immunoglobulin-like N-terminal beta-sandwich domain of unknown function, not present in the *E. coli* K-12 McrC homolog, similar to a wide range of folds from this superfamily with diverse roles (*Figure 5*, *Figure 5—figure supplements 1–4*, *Supplementary file 1*; *Anonymous, 2015*; *Halaby et al., 1999*). It is also present in non-CoCoNuT McrC homologs associated with McrB GTPase homologs in the NxD clade, implying its function is not specific to the CoCoNuTs. In the Type I-B and I-C CoCoNuT CnuC homologs, these domains most closely resemble Rho GDP-dissociation inhibitor 1, suggesting that they may be involved in the regulation of CnuB/McrB GTPase activity (*Dovas and Couchman, 2005*).

We also detected a close relative of Type I-B CoCoNuT in many *Bacillus* species, which we denoted Pseudo-Type I-B CoCoNuT because it lost the separate coiled-coil protein CnuA (*Figure 3*). We hypothesize that Pseudo-Type I-B CoCoNuTs play a role in overcrowding-induced stress responses. We inferred this functional prediction from the fact that the islands in which they occur typically also encode a quorum-sensing hormone synthase, a mechanosensitive ion channel, various transporters, antibiotic resistance and synthesis factors, and cell wall-related proteins (*Figure 3—figure supplement 2*). In many cases, we detected Type I-B CoCoNuT homologs in the extended neighborhoods of the Pseudo-Type I-B CoCoNuTs, although only rarely in the immediate vicinity. Thus, Pseudo-Type I-B CoCoNuTs might be derived duplicates of Type I-B CoCoNuTs that acquired a specialized but likely related functionality, perhaps still using the coiled-coil protein encoded by the Type I-B CoCoNuT.

## Type II and III CoCoNuT systems

Type II and III CoCoNuT CnuB/McrB GTPase domains branch from within Type I-A and are encoded in a nearly completely conserved genomic association with a Superfamily 1 (SF1) helicase of the UPF1-like clade, which we denote CnuH (*Figure 3*, *Supplementary file 1*; *Gorbalenya and Koonin, 1993*; *Fairman-Williams et al., 2010*). The UPF1-like family encompasses helicases with diverse functions acting on RNA and single-stranded DNA (ssDNA) substrates, and notably, the prototypical UPF1 RNA helicase and its closest relatives are highly conserved in eukaryotes, where they play a critical role in the nonsense-mediated decay (NMD) RNA surveillance pathway (*Cheng et al., 2007*; *Chakrabarti et al., 2011*).

SF1 helicases are composed of two RecA-like domains, which together harbor a series of signature motifs required for the ATPase and helicase activities, including the Walker A and Walker B motifs conserved in P-loop NTPases, which are located in the N-terminal RecA-like domain (*Fairman-Williams et al., 2010*). In all four Type II and Type III CoCoNuT subtypes, following the Walker A motif, the CnuH helicases contain a large helical insertion, with some of the helices predicted to form coiled-coils (*Figure 6*, *Figure 6—figure supplement 1*). The Type II, Type III-A, and Type III-B CoCoNuT CnuH helicases contain an OB-fold domain that, in Type II and Type III-A, is flanked by helices predicted to form a stalk-like helical extension of the N-terminal RecA-like domain, a structural feature characteristic of the entire UPF1/DNA2-like helicase family within SF1 (*Chakrabarti et al., 2011*; *Zhou et al., 2015*; *Kalathiya et al., 2019*; *Figure 6*, *Figure 6—figure supplement 1*). DALI comparisons show that the CnuH-predicted OB-fold domain in Type II CoCoNuTs is similar to the OB-fold domain in UPF1 and related RNA helicases SMUBP-2 and SEN1, and this holds for Type III-A as well, although,

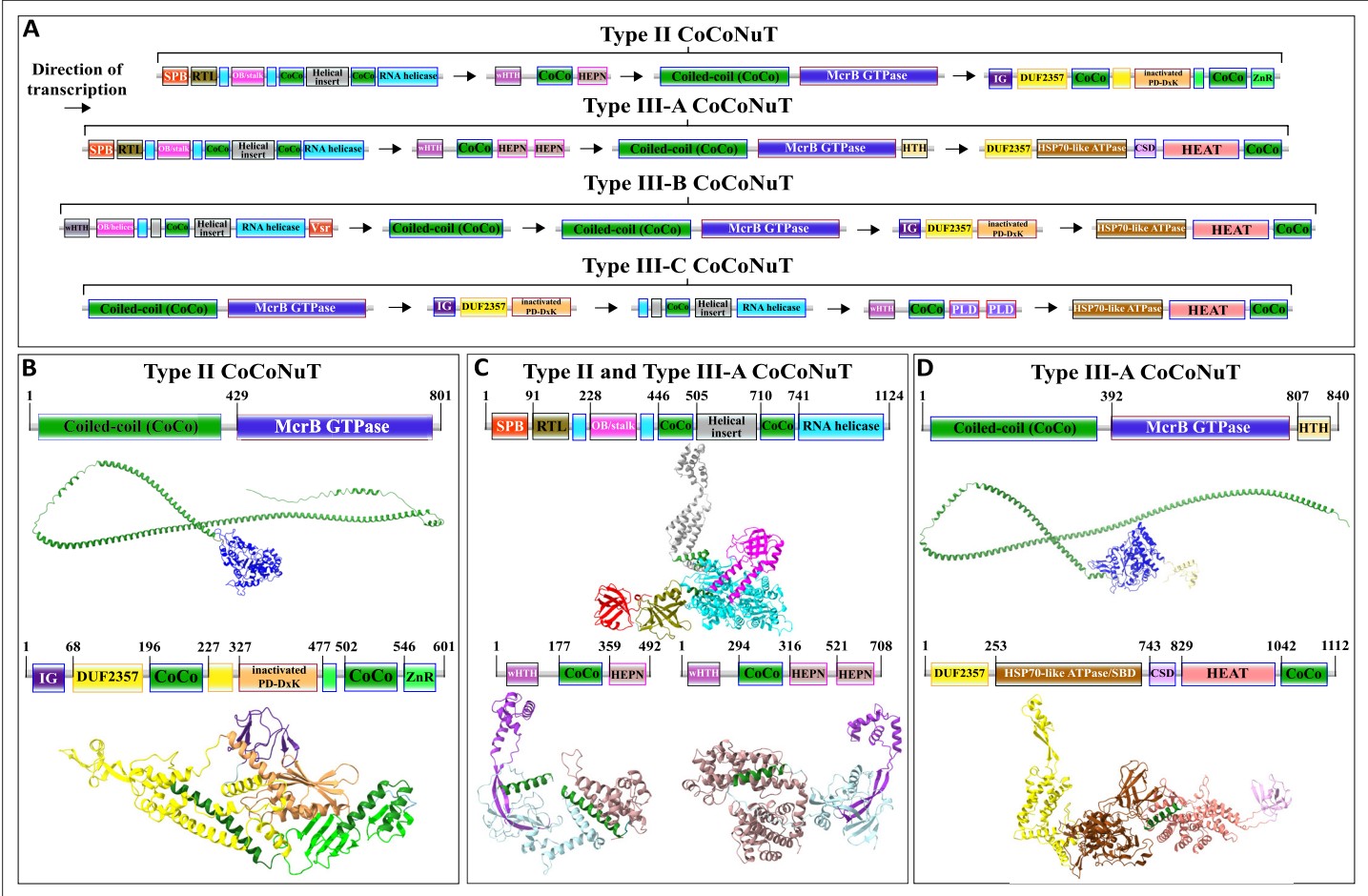

**Figure 6.** Domain composition, operon organization, and AlphaFold2 structural predictions for core protein components of Type II and III *coiled-coil nuclease tandem* (CoCoNuT) systems. (**A**) Type II and III CoCoNuT domain composition and operon organization. The arrows indicate the direction of transcription. Type II and III-A CoCoNuT systems very frequently contain TerY-P systems as well, but not invariably, and these are never found in Type III-B or III-C, thus, we do not consider them core components. (**B–D**) High-quality (average predicted local distance difference test [pLDDT] > 80), representative AlphaFold2 structural predictions for protein monomers in (**B**) Type II CoCoNuT systems (CnuB and CnuC, from top to bottom), (**C**) Type II and Type III-A CoCoNuT systems (CnuH at the top, Type II CnuE on the bottom left, Type III-A CnuE on the bottom right), and (**D**) Type III-A CoCoNuT systems (CnuB and CnuC, from top to bottom). Models were generated from representative sequences with the following GenBank accessions (see *Supplementary file 3* for sequences and locus tags): AMO81401.1 (Type II CoCoNuT CnuB), AVE71177.1 (Type II CoCoNuT CnuC), AMO81399.1 (Type II and III-A CoCoNuT CnuH), AVE71179.1 (Type II CoCoNuT CnuE), ATV59464.1 (Type III-A CnuE), PNG83940.1 (Type III-A CoCoNuT CnuB), and NMY00740.1 (Type III-A CoCoNuT CnuC). Abbreviations of domains: CSD, cold shock domain; CoCo, coiled-coil; IG, immunoglobulin (IG)-like beta-sandwich domain; ZnR, zinc ribbon domain; SPB, SmpB-like domain; RTL, RNase toxin-like domain; HEPN, HEPN family nuclease domain; OB/stalk, OB-fold domain attached to a helical stalk-like extension of ATPase; Vsr, very-short-patch-repair PD-(D/E)xK nuclease-like domain; PLD, phospholipase D family nuclease domain; HEAT, HEAT-like helical repeats. These structures were visualized with ChimeraX (*Pettersen et al., 2021*).

The online version of this article includes the following figure supplement(s) for figure 6:

**Figure supplement 1.** Domain composition and AlphaFold2 structural predictions for core protein components of Type II and III-A *coiled-coil nuclease tandem* (CoCoNuT) systems colored by predicted local distance difference test (pLDDT).

**Figure supplement 2.** AlphaFold2 prediction of Type II *coiled-coil nuclease tandem* (CoCoNuT) CnuB GTPase hexamer and CnuC monomer complex.

**Figure supplement 3.** AlphaFold2 prediction of Type II *coiled-coil nuclease tandem* (CoCoNuT) CnuH helicase and CnuE effector complex.

**Figure supplement 4.** AlphaFold2 prediction of Type III-A *coiled-coil nuclease tandem* (CoCoNuT) CnuB GTPase hexamer and CnuC monomer complex.

**Figure supplement 5.** Comparisons of Type II and Type III-A *coiled-coil nuclease tandem* (CoCoNuT) N-terminal SPB domains, SmpB, and prokaryotic HIRAN domains.

**Figure supplement 6.** Alignment of Hsp70-like nucleotide-binding domains (NBDs) from Type III *coiled-coil nuclease tandem* (CoCoNuT) and related McrB homolog with *E. coli* Hsp70 (DnaK) and mammalian Hsp70 cognate protein *H. sapiens* HSC70.

in these systems, the best DALI hits are to translation factor components such as EF-Tu domain II (*Supplementary file 1*; *Chakrabarti et al., 2011*; *Morse et al., 2020*). The Type III-B CnuH-predicted OB-folds also match that of UPF1, albeit with lower statistical support (*Supplementary file 1*).

In Type II and Type III-A CoCoNuTs, CnuH is fused at the N-terminus to a second OB-fold domain similar to that of SmpB (*s*mall *p*rotein B) (*Figure 6*, *Figure 6—figure supplements 1 and*, *5*, *Supplementary file 1*), which we denote SPB (*SmpB*-like). SmpB binds to SsrA RNA, also known as transfer-messenger RNA (tmRNA), and is required for tmRNA to rescue stalled ribosomes, via entry into their A-sites with its alanine-charged tRNA-like domain (*Barends et al., 2001*; *Himeno et al., 2014*; *Guyomar et al., 2021*). The SPB domain is around 20 amino acids shorter on average than SmpB itself, and a helix that is conserved in SmpB orthologs and interacts with the tmRNA is absent in the CoCoNuT OB folds (*Figure 6—figure supplement 5*; *Bessho et al., 2007*). However, two other structural elements of SmpB involved in binding tmRNA are present (*Figure 6—figure supplement 5*; *Gutmann et al., 2003*). The N-terminal OB-folds in Type II and III-A CnuH homologs also resemble prokaryotic HIRAN domains, which are uncharacterized, but have eukaryotic homologs fused to helicases that bind ssDNA (*Figure 6—figure supplement 5*; *Chavez et al., 2018*). These HIRAN domains, however, do not overlay with the CoCoNuT domains any better than SmpB (*Figure 6—figure supplement 5*), lack a small helix that SmpB and the CoCoNuT domains share, and likely being DNA-binding, do not fit the other pieces of evidence we gathered that all suggest an RNA-binding role for this domain in the CoCoNuTs (see below).

The Type II and Type III-A CoCoNuT CnuH helicases contain an additional domain, structurally similar to the RelE/Colicin D RNase fold (*Gucinski et al., 2019*), which we denote *R*Nase *t*oxin-*l*ike (RTL), located between the SPB OB-fold and the helicase (*Figure 6*, *Figure 6—figure supplement 1*, *Supplementary file 1*). The proteins of this family are ribosome-dependent toxins that cleave either mRNA or tRNA in the ribosomal A-site (*Pedersen et al., 2003*). This domain was identified by structural similarity search with the AF2 models, but no sequence conservation with characterized members of this family was detected, leaving it uncertain whether the RTL domain is an active nuclease. However, considerable divergence in sequence is not unusual in this toxin family (*Guglielmini and Van Melderen, 2011*; *Goeders et al., 2013*). In Type III-B CoCoNuTs, a wHTH domain resembling the archaeal ssDNA-binding protein Sul7s is fused to the CnuH N-termini (*Figure 6*, *Supplementary file 1*).

We searched for additional homologs of CnuH (see 'Methods'). Our observations of the contextual associations, both of the CoCoNuTs and their relatives, showed that helicases of this large family are typically encoded in operons with downstream genes coding for an elongated wHTH domain fused at its C-terminus to a variety of effectors, generally, nucleases, which we denote CnuE (*Figures 3 and 6*, *Figure 6—figure supplement 1*). In the CoCoNuTs, except for Types III-B and III-C, these CnuE effectors are HEPN ribonucleases, with one HEPN domain in Type II and two in Type III-A (*Figure 6*, *Figure 6—figure supplement 1*, *Supplementary file 1*). In Type III-B, the effector is a Vsr (*v*ery-short-patch *r*epair)-like PD-(D/E)xK family endonuclease (*Tsutakawa et al., 1999*) fused directly to the helicase, with no wHTH domain present, whereas in Type III-C, a distorted version of the elongated wHTH domain is fused to two phospholipase D (PLD) family endonuclease domains (*Figure 6*, *Supplementary file 1*). All these nucleases can degrade RNA, and some, such as HEPN, have been found to cleave RNA exclusively (*Pillon et al., 2021*; *Ipsaro et al., 2012*; *Mendez et al., 2018*; *Laganeckas et al., 2011*; *Songailiene et al., 2020*). We also detected coiled-coils in the region between the wHTH and effector domains in Type II, Type III-A, and Type III-C CoCoNuT CnuE homologs. Multimer structural modeling with AF2 suggests that the wHTH domain in CnuE might interact with the coiled-coil-containing helical insertion of CnuH, perhaps mediated by the coiled-coils in each protein, to couple the ATP-driven helicase activity to the various nuclease effectors (*Figure 6—figure supplement 3*). The accuracy of this model notwithstanding, the fusion of the Vsr-like effector to CnuH in Type III-B CoCoNuTs and the similarity of this system to the other CoCoNuT types strongly suggests that the CnuE effector proteins in these systems form complexes with their respective CnuH helicases. An additional factor in potential complexing by these proteins is the presence of coiled-coils in the associated CnuB/McrB and CnuC/McrC homologs, which may interact with the coiled-coils in CnuH and CnuE. Type III-B CoCoNuTs also code for a separate coiled-coil protein, which we denote CnuA, as it resembles the CnuA protein encoded in Type I-B CoCoNuTs (*Figures 3, 5 and 6*, *Figure 5—figure supplement 1*). However, it is distinguished from Type I-B CnuA in containing no recognizable

domains other than the coiled-coil (*Figures 3, 5, and 6*, *Figure 5—figure supplement 1*). This also could potentially interact with other coiled-coil proteins in the system.

Type II and Type III-A CoCoNuTs, the most widespread varieties apart from Type I, also include conserved genes coding for a 'TerY-P' triad. TerY-P consists of a TerY-like von Willebrand factor type A (VWA) domain, a protein phosphatase 2C-like enzyme, and a serine/threonine kinase (STK) fused at the C-terminus to a zinc ribbon (ZnR) (*Figure 3*, *Supplementary file 1*). TerY-P triads are involved in tellurite resistance, associated with various predicted DNA restriction and processing systems, and are hypothesized to function as a metal-sensing phosphorylation-dependent signaling switch (*Anantharaman et al., 2012*). In addition, TerY-P-like modules, in which the kinase is fused at the C-terminus to an OB-fold rather than a zinc ribbon, have been recently shown to function as stand-alone antiphage defense systems (*Gao et al., 2020*). The OB-fold fusion suggests that this kinase interacts with an oligonucleotide and raises the possibility that the zinc ribbon, which occupies the same position in the CoCoNuTs, is also nucleic acid-binding. Almost all CoCoNuT systems containing *cnuHE* operons also encompass TerY-P, with a few exceptions among Terrabacterial Type II systems and Myxococcal Type III-A systems, implying an important contribution to their function (*Supplementary file 3*). However, the complete absence of the TerY-P module in Type III-B and Type III-C systems suggests that when different nucleases and other putative effectors fused to the helicase are present, TerY-P is dispensable for the CoCoNuT activity. Therefore, we do not consider them to be core components of these systems.

Type II and Type III-A CoCoNuTs have similar domain compositions, but a more detailed comparison reveals substantial differences. The CnuE proteins in Type II contain one HEPN domain with the typical RxxxxH RNase motif conserved in most cases, whereas those in Type III-A contain two HEPN domains, one with the RxxxxH motif, and the other, closest to the C-terminus, with a shortened RxH motif. Furthermore, Type III-A CnuB/McrB GTPase homologs often contain C-terminal HTH-domain fusions absent in Type II (*Figure 6*, *Figure 6—figure supplements 1 and 4*). Finally, striking divergence has occurred between the Type II and III-A CnuC/McrC homologs. Type II CnuCs resemble Type I-A CnuCs, which contain N-terminal immunoglobulin-like beta-sandwich domains, PD-(D/E)xK nucleases, zinc ribbon domains, and insertions into DUF2357 containing coiled-coils and a CSD-like OBD. However, in Type II CnuCs, this domain architecture underwent reductive evolution (*Figure 5*, *Figure 5—figure supplements 1 and 2*, *Figure 6*, *Figure 6—figure supplements 1 and 2*, *Supplementary file 1*). In particular, the beta-sandwich domain and coiled-coils are shorter, the CSD was lost, the nuclease domain was inactivated, and, in many cases, the number of Zn-binding CPxC motifs was reduced from three to two (*Figure 5*, *Figure 5—figure supplements 1 and 2*, *Figure 6*, *Figure 6—figure supplements 1 and 2*, *Supplementary file 1*). This degeneration pattern could indicate functional replacement by the associated CnuH and CnuE proteins, often encoded in reading frames overlapping with the start of the *cnuBC* operon.

By contrast, Type III-A CnuC/McrC homologs entirely lost the beta-sandwich domain, PD-(D/E)xK nuclease, and zinc ribbon found in Type I-A and Type II, but gained an Hsp70-like NBD/SBD unit similar to those fused to the McrB-like GTPase domain in early branching members of the NxD clade. They have also acquired a helical domain similar to the HEAT repeat family, and, in some cases, a second CSD (*Figures 3 and 5*, *Figure 5—figure supplement 1*, *Figure 6*, *Figure 6—figure supplements 1 and 4*, *Supplementary file 1*). Most Type III-A CnuCs contain a CSD and coiled-coils, and thus, resemble Type I-A, but the positioning of these domains, which in Type I-A are inserted into the DUF2357 helix bundle, is not conserved in Type III-A, where these domains are located outside DUF2357 (*Figure 5*, *Figure 5—figure supplement 1*, *Figure 6*, *Figure 6—figure supplements 1 and 4*).

Hsp70-like ATPase NBD/SBD domains and HEAT-like repeats are fused to the CnuC/McrC N-terminal DUF2357 domain in Type III-A CoCoNuT, but their homologs in Types III-B and III-C are encoded by a separate gene. We denote these proteins CnuD and CnuCD, the latter for the CnuC-CnuD fusions in Type III-A. The separation of these domains in Types III-B and III-C implies that fusion is not required for their functional interaction with the CnuBC/McrBC systems (*Figures 3 and 6A*, *Supplementary file 1*). CnuD proteins associated with both Types III-B and III-C usually contain predicted coiled-coils, suggesting that they might interact with the large coiled-coil in the CnuB homologs (*Figure 6A*).

In the CnuD homologs found in the CoCoNuTs and fused to NxD McrB GTPases, Walker B-like motifs (*Yamamoto et al., 2014*) are usually, but not invariably, conserved, whereas the sequences of

the helical domains adjacent to the motifs are more strongly constrained. Walker A-like motifs (*Chang et al., 2008*) are present but degenerate (*Figure 6—figure supplement 6*). Therefore, it appears likely that the CoCoNuT CnuD homologs bind ATP/ADP but hydrolyze ATP with extremely low efficiency, at best. Such properties in an Hsp70-like domain are better compatible with RNA binding than unfolded protein binding or remodeling, suggesting that these CnuD homologs may target the respective systems to viral/aberrant RNA. Many Hsp70 homologs have been reported to associate with ribosomes (*Mayer, 2021*; *Willmund et al., 2013*), which could also be true for the CoCoNuT Cnu(C)Ds.

Consistent with this prediction, Type II and III-A CoCoNuTs likely target RNA rather than DNA, given that the respective operons encode HEPN RNases, typically the only recognizable nuclease in these systems. All CnuC/McrC homologs in Type II or III CoCoNuT systems lack the PD-(D/E)xK catalytic motif that is required for nuclease activity, although for Type II and Types III-B and III-C, but not Type III-A, structural modeling indicates that the inactivated nuclease domain was retained, likely for a nucleic acid-binding role (*Figure 6*, *Supplementary file 1*). In Type III-A CoCoNuT, the nuclease domain was lost entirely and replaced by the Hsp70-like NBD/SBD domain with RNA-binding potential described above (*Figure 6*, *Figure 6—figure supplements 1 and 4*, *Supplementary file 1*). Often, one or two CSDs, generally RNA-binding domains, although capable of binding ssDNA, are fused to Type III-A CnuC homologs as well (*Figure 6*, *Figure 6—figure supplements 1 and 4*, *Supplementary file 1*; *Heinemann and Roske, 2021*). RNA targeting capability of Types III-B and III-C can perhaps be inferred from their similarity to Type III-A in encoding CnuD Hsp70-like proteins. Moreover, higher-order associations of Type II and Type III-A CoCoNuT systems with various DNA restriction systems suggest a two-pronged DNA and RNA restriction strategy reminiscent of Type III CRISPR-Cas (see below).

We suspect that RNA targeting is an ancestral feature of the CoCoNuT systems. Several observations are compatible with these hypotheses:

1. Most of the CoCoNuTs encompass HEPN nucleases that appear to possess exclusive specificity for RNA.
2. CSD-like OB-folds are pervasive in these systems, being present in the CnuB/McrB homologs of all Type I subtypes and in Type I-A and Type III-A CnuC/McrC homologs. As previously noted, these domains typically bind RNA, although they could bind ssDNA as well.
3. YTH-like domains are present in most Type I CoCoNuTs, particularly, in almost all early branching Type I-B and I-C systems, suggesting that the common ancestor of the CoCoNuTs contained such a domain. YTH domains in eukaryotes sense internal N6-methyladenosine (m6A) in mRNA (*Hazra et al., 2019*; *Liao et al., 2018*; *Patil et al., 2018*).
4. Type II and Type III-B/III-C CoCoNuTs, which likely target RNA, given the presence of HEPN domains and Hsp70 NBD/SBD homologs, retain inactivated PD-(D/E)xK nuclease domains, suggesting that these domains contribute an affinity for RNA inherited from Type I-A CoCoNuTs. PD-(D/E)xK nucleases are generally DNA-specific; however, some examples of RNase activity have been reported (*Mendez et al., 2018*; *Laganeckas et al., 2011*). The inactivated PD-(D/E)xK domains might also bind DNA from which the target RNA is transcribed.

## Extension of Type III-A CoCoNuT systems with ATPases and virulence factors

We observed even greater levels of complexity in Type III-A CoCoNuTs, which might ultimately beget another level of classification, where the TerY-related VWA domains were duplicated. In each of the three subtypes of Type III-A, a different ATPase was inserted between these VWA domains into the predicted operon. In these systems, the $Mg^{2+}$-coordinating MIDAS motif (DxSxS…T...D) is perfectly conserved in the VWA domain encoded at the 5′ end of the operon, which resembles the domains found in systems with only one VWA domain, whereas the internal VWA domain lacks the middle threonine and is slightly shorter (*Figure 7*, *Supplementary file 1*; *Lacy et al., 2004*). In the first two of these Type III-A subtypes shown in *Figure 7*, a CARF (*CRISPR-Associated Rossmann Fold*) domain-containing protein is encoded, with two distinct CARF domains encompassing different RING nuclease motifs. These CARF domains are fused at the C-terminus to a D-ExK nuclease domain, an architecture suggestive of a PCD/dormancy-eliciting antiphage effector (*Supplementary file 1*; *Makarova et al., 2014*; *Makarova et al., 2020a*). Indeed, homologs of this protein, Can1 and Can2, have been

**Figure 7.** Compound Type III-A *coiled-coil nuclease tandem* (CoCoNuT) operons with 5' extensions. Additional genes that may be present in Type III-A CoCoNuT operons. Abbreviations of domains: McrB, McrB-like GTPase domain; CoCo/CC, coiled-coil; STK, serine/threonine kinase; 2xCARF, 2 CARF domains; D-ExK, D-ExK nuclease motif; MN, McrC N-terminal domain (DUF2357); CSD, cold shock domain; ZnR, zinc ribbon domain; SPB, SmpB-like domain; RTL, RNase toxin-like domain; OB, OB-fold domain attached to helical stalk-like extension of ATPase; HEPN, HEPN family nuclease domain; Hsp70, Hsp70-like NBD/SBD; HEAT, HEAT-like helical repeats; LRR, leucine-rich repeat; Gly_zip, glycine zipper domain; SpoVK, EssC, EccE3-HerA – see text.

characterized as CRISPR ancillary nucleases (*McMahon et al., 2020*; *Zhu et al., 2021*), and Can2 has been reported to cleave both DNA and RNA (*Zhu et al., 2021*). In these two CoCoNuT varieties, the CnuC/McrC homologs typically contain two CSDs, whereas in the third type, where the CARF proteins are not encoded, there is a single CSD (*Figure 7*, *Supplementary file 1*).

The first of the three variable regions flanked by the VWA domain-encoding genes includes an FtsK-like ATPase homologous to the Type VII secretion system factor EssC (*Figure 7*, *Supplementary file 1*). EssC contains three tandem ATPase domains (D1, D2, and D3), with the Walker A/B motifs required for ATP hydrolysis present only in D1 and D2 (*Warne et al., 2016*; *Bobrovskyy et al., 2022*). The CoCoNuT-associated homologs also possess three ATPase domains but differ in that only the central D2 domain homolog is predicted to be active. They are also distinguished from EssC by the presence of a coiled-coil that can exceed 200 residues in length and is fused at their N-terminus, whereas forkhead-associated domains and transmembrane helices are found in this position in EssC (*Supplementary file 1*; *Bobrovskyy et al., 2022*; *Warne et al., 2016*). In addition, the region codes for two WXG100 proteins, one with the characteristic WxG motif and the other without (but with a similar predicted structure), as well as a coiled-coil fused to a restriction endonuclease-like domain with a D-ExK catalytic motif (*Figure 7*, *Supplementary file 1*; *Poulsen et al., 2014*). Finally, another protein similar to DNA mimics that bind the HU histone-like factor is encoded following the coiled-coil-nuclease fusion (*Figure 7*, *Supplementary file 1*; *Wang et al., 2013*).

It appears likely that some or all of these factors are secreted, especially the WXG100 proteins, which are known to be secreted, with the prototypical example, ESAT-6, being a T-cell antigen diagnostic of *Mycobacterium tuberculosis* infection (*Poulsen et al., 2014*). These associated WXG100 proteins implicate the ATPases in defensive protein secretion, but as they lack transmembrane domains present in their EssC homologs, a different mechanism appears likely. The presence of coiled-coils at the N-termini of these proteins suggests that they might interact with the coiled-coils in the core CoCoNuT factors and/or with the associated coiled-coil-nuclease fusion. The FtsK superfamily ATPases form hexamers (*Bobrovskyy et al., 2022*), so should such an interaction occur, they are likely compatible with the CnuB/McrB GTPase hexamer. A prior study that tangentially examined these operons in the context of the TerY-P triad pointed out that this WXG100/FtsK-like ATPase operon is likely a mobile element that can be found as a stand-alone secretion system in other genomes (*Anantharaman et al., 2012*).

In the second of these Type III-A CoCoNuT extensions with two VWA domains, a SpoVK family of AAA+ATPases homologous to p97/CDC48 is encoded adjacent to a protein containing a C-terminal bacteriocin-like glycine zipper motif (*Figure 7*, *Supplementary file 1*). CDC48 is involved in eukaryotic protein quality control, particularly the degradation of proteins synthesized from non-stop mRNA,

where it is required to release nascent polypeptides from stalled ribosomes to enable proteolysis (*Verma et al., 2013*). CDC48 contains two tandem ATPase domains, both of which are active; the CoCoNuT-associated homologs also contain tandem ATPases, but the Walker A/B motifs required to bind and hydrolyze ATP are conserved only in the C-terminal domain (*Baek et al., 2013*; *Wolf and Stolz, 2012*). As members of the AAA+ superfamily, these ATPases assemble into hexamers (*Wolf and Stolz, 2012*), similarly to the McrB family GTPases. At their C-termini, these ATPases are fused to domains of unknown function, namely, a leucine-rich repeat element and a beta-barrel domain structurally similar to biotin carrier proteins, suggesting that these proteins might be biotinylated (*Figure 7*, *Supplementary file 1*; *Choi-Rhee and Cronan, 2003*). The conserved association of the CDC48-like ATPases with these Type III-A CoCoNuTs, which encode the potentially tmRNA-binding SPB domain, seems to provide support for the scenario of tmRNA interaction. Structural analysis of the bacteriocin-like protein encoded in these loci indicates that it adopts an inactivated PD-(D/E) xK-type restriction endonuclease fold, potentially nucleic acid-binding (*Supplementary file 1*). These proteins might be analogous to the restriction endonuclease-like factors in the FtsK/EssC homolog neighborhoods (*Figure 7*).

The third variant of these extended Type III-A CoCoNuTs encodes a distinct member of the FtsK/ HerA superfamily, which is also likely assembled into hexamers (*Figure 7*, *Supplementary file 1*; *Iyer et al., 2004b*). AF2 structural modeling suggests these enzymes are homologs of the ESX-3 Type VII secretion system factor EccE3 (*Poweleit et al., 2019*), albeit containing a unique beta-strand insertion of variable length. This EccE3-like domain is fused to an ATPase domain similar to the Type IV secretion system protein VirB4, which is involved in bacterial conjugation (*Figure 7*, *Supplementary file 1*; *Wallden et al., 2010*). These systems also encode a small helical domain of unknown function in operonic association with the ATPase (*Figure 7*). These genes might be involved in the mobilization of the locus via conjugation or instead play a similar role in secretion as predicted for the EssC-like ATPases. However, WXG100 homologs, like those that strongly imply a secretion-related function for the EssC-like ATPases, are not encoded near these VirB4-like ATPases. Lastly, we observed that, unlike the EssC-like ATPases and the SpoVK-like proteins, these enzymes are not always encoded between VWA genes at the 5′ end of the operons, but in some cases, migrated to the 3′ end; in these cases, however, duplicated VWA domains are present at the 5′ end, a potential vestige of an extension that was recently lost or relocated.

Overall, these elaborations of Type III-A CoCoNuT systems resemble the TerY-P triads in that they could be stand-alone defensive cassettes that augment the effectiveness of the core CoCoNuT systems. It is unclear, however, why these types of factors are flanked by TerY-like VWA domains, as opposed to restriction systems such as Type I RM, GmrSD, and Druantia Type III, which are commonly associated with Type III-A CoCoNuTs as well, but are never so tightly integrated into the operon (*Loenen et al., 2014*; *Weigele and Raleigh, 2016*; *Doron et al., 2018*). Type III-A systems embedded in these extended operons are annotated in *Supplementary file 3*.

While investigating this additional diversity of Type III-A CoCoNuTs, we observed that Type III-A CoCoNuTs in *Helicobacter* appeared to be translated using an alternate genetic code because gene predictions with the standard code divided the expected open-reading frames into many small fragments. We were unable to identify a known alternative code that would yield the expected CoCoNuT gene products. Thus, a novel type of conditional or otherwise complex translation regulation likely occurs in these species, perhaps triggered by phage infection (for the accessions of identifiable Type III-A CoCoNuT factors in *Helicobacter*, see *Supplementary file 3*).

## Complex higher-order associations between CoCoNuTs, CARF domains, and other defense systems

Genomic neighborhoods of many Type II and Type III-A CoCoNuTs encompass complex operonic associations with genes encoding several types of CARF domain-containing proteins (*Figures 7 and 8*, *Figure 8—figure supplement 1*). This connection suggests multifarious regulation by cyclic (oligo) nucleotide second messengers synthesized in response to viral infection and bound by CARF domains (*Makarova et al., 2020a*; *McMahon et al., 2020*; *Zhu et al., 2021*). Activation of an effector, most often a nuclease, such as HEPN or PD-(D/E)xK, by a CARF bound to a cyclic (oligo)nucleotide is a crucial mechanism of CBASS (*cyclic oligonucleotide-based antiphage signaling system*) as well as Type III CRISPR-Cas systems (*McMahon et al., 2020*; *Makarova et al., 2020a*; *Zhu et al., 2021*). These

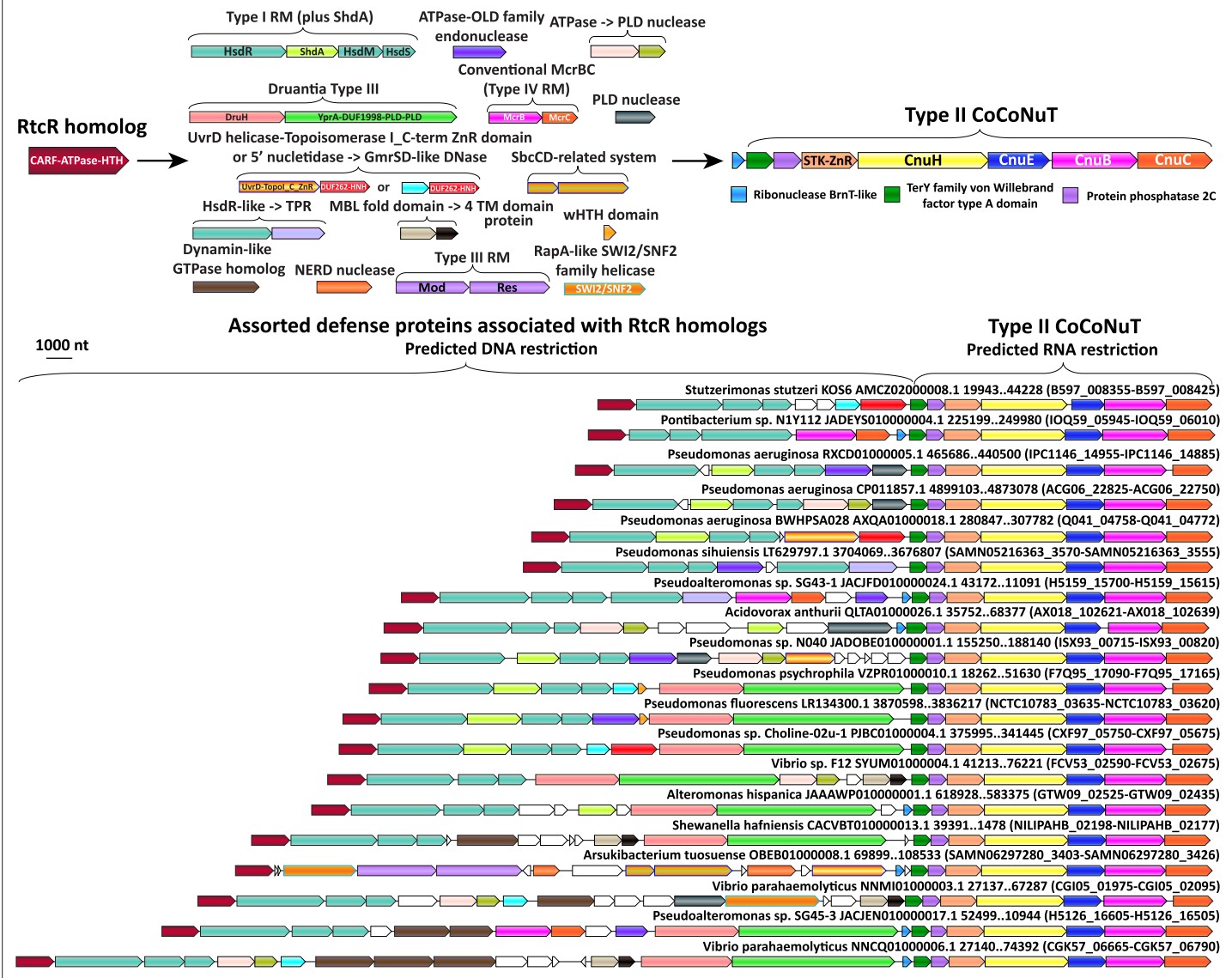

**Figure 8.** Complex operonic associations of Type II *coiled-coil nuclease tandems* (CoCoNuTs). Type II CoCoNuTs are frequently associated with RtcR homologs, and in many cases, ancillary defense genes are located between the RtcR gene and the CoCoNuT, almost always oriented in the same direction in an apparent superoperon. Abbreviations of domains: STK, serine/threonine kinase; ZnR, zinc ribbon domain; YprA, YprA-like helicase domain; DUF1998, DUF1998 is often found in or associated with helicases and contains four conserved, putatively metal ion-binding cysteine residues; PLD, phospholipase D family nuclease domain; SWI2/SNF2, SWI2/SNF2-family ATPase; HsdR/M/S, Type I RM system restriction, methylation, and specificity factors; ShdA, shield system core component ShdA; TPR, tetratricopeptide repeat protein; MBL fold, metallo-beta-lactamase fold; 4 TM domain, protein with four predicted transmembrane helices; Mod/Res, Type III RM modification and restriction factors.

The online version of this article includes the following figure supplement(s) for figure 8:

**Figure supplement 1.** Type II *coiled-coil nuclease tandems* (CoCoNuTs) are associated with RtcR homologs in a variety of species.

CARF-regulated enzymes generally function as a fail-safe that eventually induces PCD/dormancy when other antiphage defenses fail to bring the infection under control and are deactivated, typically through cleavage of the second messenger by a RING nuclease, if other mechanisms succeed (*Makarova et al., 2012*; *Koonin and Zhang, 2017*; *Makarova et al., 2020a*; *Koonin and Krupovic, 2019*).

The presence of CARFs could implicate the HEPN domains of these systems as PCD effectors that would carry out non-specific RNA degradation in response to infection. Surprisingly, however, most of these CARF domain-containing proteins showed the highest similarity to RtcR, a sigma54 transcriptional coactivator of the RNA repair system RtcAB with a CARF-ATPase-HTH domain architecture,

suggesting an alternative functional prediction (*Figure 8*, *Figure 8—figure supplement 1*, *Supplementary file 1*). Specifically, by analogy with RtcR, CoCoNuT-associated CARF domain-containing proteins might bind (t)RNA fragments with 2′,3′ cyclic phosphate ends (*Kotta-Loizou et al., 2022*; *Hughes et al., 2020*). This interaction could promote transcription of downstream genes, in this case, genes encoding CoCoNuT components, through binding an upstream activating sequence by the HTH domain fused to the CARF-ATPase C-terminus (*Hughes et al., 2020*; *Kotta-Loizou et al., 2022*).

The manifold biological effects of tRNA-like fragments are only beginning to be appreciated. Lately, it has been shown that bacterial anticodon nucleases, in response to infection and DNA degradation by phages, generate tRNA fragments, likely a signal of infection and a defensive strategy to slow down the translation of viral mRNA, and that phages can deploy tRNA repair enzymes and other strategies to counteract this defense mechanism (*Bitton et al., 2015*; *van den Berg et al., 2023*; *Kaufmann, 2000*; *Ishita et al., 2021*). Moreover, the activity of the HEPN ribonucleases in the CoCoNuTs themselves would produce RNA cleavage products with cyclic phosphate ends that might be bound by the associated RtcR-like CARFs (*Shigematsu et al., 2018*; *Pillon et al., 2021*), in a potential feedback loop.

Such a CoCoNuT mechanism could complement the function of RtcAB as RtcA is an RNA cyclase that converts 3′-phosphate RNA termini to 2′,3′-cyclic phosphate and thus, in a feedback loop, generates 2′,3′-cyclic phosphate RNA fragments that induce expression of the operon (*Genschik et al., 1998*; *Das and Shuman, 2013*; *Hughes et al., 2020*). Acting downstream of RtcA, RtcB is an RNA ligase that joins 2′,3′-cyclic phosphate RNA termini to 5′-OH RNA fragments, generating a 5′–3′ bond and, in many cases, reconstituting a functional tRNA (*Tanaka and Shuman, 2011*). Recent work has shown that, in bacteria, the most frequent target of RtcB is SsrA, the tmRNA (*Kotta-Loizou et al., 2022*). Intriguingly, as described above, in order to rescue stalled ribosomes, tmRNA must bind to SmpB, an OB-fold protein highly similar to the predicted structures of the SPB domains fused at the N-termini of the CnuH helicases in Type II and III-A CoCoNuTs, which are the only types that frequently associate with RtcR homologs (*Figure 8*, *Figure 8—figure supplement 1*; *Himeno et al., 2014*; *Guyomar et al., 2021*).

There are notable parallels between CoCoNuTs and Type III CRISPR-Cas systems, where the Cas10-Csm-crRNA effector complex binds phage RNA complementary to the spacer of the crRNA, triggering both restriction of phage DNA and indiscriminate cleavage of RNA (*McMahon et al., 2020*). The target RNA recognition stimulates the production of cyclic oligoadenylate (cOA) signal molecules by Cas10, and these bind the CARF domain of PCD effectors, such as Csm6, activating their nuclease moieties, typically HEPN domains that function as promiscuous RNases (*Pillon et al., 2021*; *Millman et al., 2020*). One of the two outcomes can result from this cascade: infection is either eradicated quickly by restriction of the virus DNA, which inhibits cOA signaling via the depletion of viral RNA, along with the activity of RING nucleases, thus averting PCD, or else, the continued presence of viral RNA stimulates cOA signaling until PCD or dormancy occurs, limiting the spread of viruses to neighboring cells in the bacterial population (*Makarova et al., 2012*; *Koonin and Aravind, 2002*; *Koonin and Krupovic, 2019*).

If CoCoNuTs associated with RtcR homologs can induce PCD, a conceptually similar but mechanistically distinct phenomenon might occur. Although many of these CoCoNuTs only contain an appended gene encoding a CARF domain-containing protein at the 5′ end of the predicted operon (*Figure 8—figure supplement 1*), there are also numerous cases where several types of DNA restriction systems are encoded between the CARF gene and the CoCoNuT (*Figure 8*). In these cases, nearly all genes are in an apparent operonic organization that can extend upward of 40 kb (*Figure 8*). Although internal RtcR-independent promoters likely exist in these large loci, the consistent directionality and close spacing of the genes in these superoperons suggests coordination of expression. The complex organization of the CARF-CoCoNuT genomic regions, and by implication, the corresponding defense mechanisms, might accomplish the same effect as Type III CRISPR-Cas, contriving a no-win situation for the target virus. Under this scenario, the virus is either destroyed by the activity of the DNA restriction systems, which would inhibit signal production (likely RNA fragments with cyclic phosphate ends rather than cOA) and drive down CoCoNuT transcription, thereby preventing PCD, or as the virus replicates, signaling and CoCoNuT transcription would continue until the infection is aborted by PCD or dormancy caused by the degradation of host mRNA by the HEPN RNase(s) of the CoCoNuT. Another noteworthy observation consistent with PCD induction is the frequent presence of

BrnT-like RelE family RNase toxins encoded in the same direction and immediately upstream of RtcR-associated CoCoNuTs (*Figure 8*, *Figure 8—figure supplement 1*, *Supplementary file 1*). A comparison of these sequences with BrnT shows notable conservation in the RNA-binding regions, but not all residues required for toxicity of BrnT are conserved (*Heaton et al., 2012*). However, as described above in reference to the RelE-like RTL domain in CnuH, nucleases of this family can vary considerably in sequence while remaining active toxins (*Guglielmini and Van Melderen, 2011*; *Goeders et al., 2013*).

In many species of *Pseudomonas*, where CoCoNuTs are almost always associated with RtcR and various ancillary factors, Type I RM systems often contain an additional gene that encodes a transmembrane helix and a long coiled-coil fused to an RmuC-like nuclease (*Figure 8*). These proteins were recently described as ShdA, the core component of the *Pseudomonas*-specific defense system Shield (*Macdonald et al., 2022*). These can potentially interact with coiled-coils in the CoCoNuTs, perhaps, guiding them to the DNA from which RNA targeted by the CoCoNuTs is being transcribed, or vice versa (*Figure 8*).

A notable difference between the generally similar, dual DNA and RNA-targeting mechanism of many Type III CRISPR-Cas systems and the proposed mechanism of the CoCoNuTs is that viral RNA recognition by the Cas10-Csm complex, rather than binding of a second messenger to a CARF domain, activates both DNA cleavage activity by the HD domain and production of cOA that triggers non-specific RNA cleavage. In contrast, in the CARF-containing CoCoNuTs, both the DNA and RNA restriction factors appear to be arranged such that binding of a 2',3' cyclic phosphate RNA fragment by the CARF domain would initiate the expression of the entire gene cluster (*McMahon et al., 2020*). In the case of the CoCoNuTs, signals of infection could promote transcription, first of the DNA restriction systems, and then, the CoCoNuT itself, a predicted RNA restriction system. In these complex configurations of the CoCoNuT genome neighborhoods (*Figure 8*), the gene order is likely to be important, with Type I RM almost always directly following CARF genes and CoCoNuTs typically coming last, although Druantia Type III sometimes follows the CoCoNuT. As translation in bacteria is co-transcriptional, the products of genes transcribed first would accumulate before those of the genes transcribed last, so that the full, potentially suicidal impact of the CoCoNuT predicted RNA nucleolytic engine would only be felt after the associated DNA restriction systems had ample time to act – and possibly, fail (*Irastortza-olaziregi and Amster-choder, 2020*).

## Conclusion

In recent years, systematic searches for defense systems in prokaryotes, primarily by analysis of defense islands, revealed enormous, previously unsuspected diversity of such systems that function through a remarkable variety of molecular mechanisms. In this work, we uncovered the hidden diversity and striking hierarchical complexity of a distinct class of defense mechanisms, the Type IV (McrBC) restriction systems. We then zeroed in on a single major but previously overlooked branch of the McrBC systems, which we denoted CoCoNuTs for their salient features, namely, the presence of extensive coiled-coil structures and tandem nucleases. Astounding complexity was discovered at this level as well, with three distinct types and multiple subtypes of CoCoNuTs that differ by their domain compositions and genomic associations. All CoCoNuTs contain domains capable of interacting with translation system components, such as the SmpB-like OB-fold, Hsp70 homologs, or YTH domains, along with RNases, such as HEPN, suggesting that at least one of the activities of these systems targets RNA. Most of the CoCoNuTs are potentially endowed with DNA-targeting activity as well, either by factors integral to the system, such as the McrC-like nuclease, or more loosely associated, such as Type I RM and Druantia Type III systems that are encoded in the same predicted superoperons with many CoCoNuTs. Numerous CoCoNuTs are associated with proteins containing CARF domains, suggesting that cyclic (oligo)nucleotides regulate the CoCoNuT activity. Given the presence of the RtcR-like CARF domains, it appears likely that the specific second messengers involved are RNA fragments with cyclic phosphate termini. We hypothesize that the CoCoNuTs, in conjunction with ancillary restriction factors, implement an echeloned defense strategy analogous to that of Type III CRISPR-Cas systems, whereby an immune response eliminating virus DNA and/or RNA is launched first, but then, if it fails, an abortive infection response leading to PCD/dormancy via host RNA cleavage takes over.

## Methods

### Comprehensive identification and phylogenetic and genomic neighborhood analysis of McrB and McrC proteins

The comprehensive search for McrB and McrC proteins was seeded with publicly available multiple sequence alignments COG1401 (McrB GTPase domain), AAA_5 (the branch of AAA+ATPases containing the McrB GTPase), COG4268 (McrC), PF10117 (McrBC), COG1700 (McrC PD-(D/E)xK nuclease domain), PF04411 (McrC PD-(D/E)xK nuclease domain), and PF09823 (McrC N-terminal DUF2357). Additional alignments and individual queries were derived from data from our previous work on the EVE domain family (*Bell et al., 2020*). All alignments were clustered, and each sub-alignment or individual query sequence was used to produce a position-specific scoring matrix (PSSM). These PSSMs were used as PSI-BLAST queries against the non-redundant (nr) NCBI database (*E*-value ≤10) (*Altschul et al., 1997*). Although a branch of McrB GTPase homologs has been described in animals, these are highly divergent in function, and no associated McrC homologs have been reported (*Iyer et al., 2004a*). Therefore, we excluded eukaryotic sequences from our analysis to focus on the composition and contextual connections of prokaryotic McrBC systems.

Genome neighborhoods for the hits were generated by downloading their gene sequence, coordinates, and directional information from GenBank, as well as for 10 genes on each side of the hit. Domains in these genes were identified using PSI-BLAST against alignments of domains in the NCBI Conserved Domain Database (CDD) and Pfam (*E*-value 0.001). Some genes were additionally analyzed with HHpred for validation of the BLAST hits or if no hits were obtained (*Gabler et al., 2020*). Then, these neighborhoods were filtered for the presence of a COG1401 hit (McrB GTPase domain), or hits to both an McrB 'alias' (MoxR, AAA_5, COG4127, DUF4357, Smc, WEMBL, Myosin_tail_1, DUF3578, EVE, Mrr_N, pfam01878) and an McrC "alias" (McrBC, McrC, PF09823, DUF2357, COG1700, PDDEXK_7, RE_LlaJI). These aliases were determined from a preliminary manual investigation of the data using HHpred (*Zimmermann et al., 2018*). Several of the McrB aliases are not specific to McrB, and instead are domains commonly fused to McrB GTPase homologs, or are larger families, such as AAA_5, that contain McrB homologs. We found that many bona fide McrB homologs, validated by HHpred, produced hits not to COG1401 but rather to these other domains, so we made an effort to retain them.

The McrB candidates identified by this filtering process were clustered to a similarity threshold of 0.5 with MMseqs2 (*Hauser et al., 2016*), after which the sequences in each cluster were aligned with MUSCLE (*Edgar, 2004*). Next, profile-to-profile similarity scores between all clusters were calculated with HHsearch (*Söding, 2005*). Clusters with high similarity, defined as a pairwise score to self-score ratio >0.1, were aligned to each other with HHalign (*Söding et al., 2006*). This procedure was performed for a total of three iterations. The alignments of each cluster resulting from this protocol, which included some false-positive clusters consisting of other members of the AAA_5 family, were analyzed with HHpred to remove the false positives, after which the GTPase domain sequences were extracted manually using HHpred, and the alignments were used as queries for a second round of PSI-BLAST against the nr NCBI database as described above. At this stage, the abundance of the CoCoNuT and CoCoPALM see above types of McrB GTPases had become apparent; therefore, results of targeted searches for these subtypes were included in the pool of hits from the second round of PSI-BLAST.

Genome neighborhoods were generated for these hits and filtered for aliases as described above. Further filtering of the data, which did not pass this initial filter, involved relaxing the criteria to include neighborhoods with only one hit to an McrB or McrC alias, but with a gene adjacent to the hit (within 90 nucleotides), oriented in the same direction as the hit, and encoding a protein of sufficient size (>200 aa for McrB, >150 aa for McrC) to be the undetected McrB or McrC component. Afterward, we filtered the remaining data to retain genome islands with no McrBC aliases but with PSI-BLAST hits in operonic association with genes of sufficient size, as described above, to be the other McrBC component. These data, which contained many false positives but captured many rare variants, were then clustered and analyzed as described above to remove false positives. Next, an automated procedure was developed to excise the GTPase domain sequences using the manual alignments generated during the first phase of the search as a reference. These GTPase sequences were further analyzed by clustering and HHpred to remove false positives.

Definitive validation by pairing McrB homologs with their respective McrC homologs was also used to corroborate their identification. Occasionally, the McrB and McrC homologs were separated by intervening genes, or the operon order was reversed, and consideration of those possibilities allowed the validation of many additional systems. The pairing process was complicated by, and drew our attention to, the frequent occurrence of multiple copies of McrBC systems in the same islands that may function cooperatively. Lastly, the rigorously validated set of McrBC pairs, supplemented only with orphans manually annotated as McrBC components using HHpred, were used for our phylogenetics. The final alignments of GTPase and DUF2357 domains were produced using the iterative alignment procedure described above for 10 iterations. Approximately maximum-likelihood trees were built with the FastTree program (*Price et al., 2010*) from representative sequences following clustering to a 0.9 similarity threshold with MMseqs2.

## Protein domain detection and annotation

Protein domains in McrBC homologs and proteins encoded by neighboring genes were initially identified using the method described above, the first pass using PSI-BLAST against alignments of domains in the NCBI CDD and Pfam (*E*-value 0.001). In many cases where no domains could be confidently detected with this method, or for validation of the hits from the first pass, HHpred was used for more sensitive analysis (*Zimmermann et al., 2018*). The CoCoNuT system components were subjected to additional scrutiny using the coiled-coil detection and visualization tool Waggawagga, which employs several algorithms for coiled-coil prediction, including Marcoil, Multicoil2, Ncoils, and Paircoil2 (*Simm et al., 2015*; *Delorenzi and Speed, 2002*; *Trigg et al., 2011*; *Lupas et al., 1991*; *McDonnell et al., 2006*). These predictions varied in their strength, with the long coiled-coils detected in CnuA and CnuB homologs having the highest likelihood (usually the maximum P-score of 100 with Marcoil and Multicoil2) and being recognized by the most of the applied tools (BLAST, HHpred, and multiple algorithms used by Waggawagga). The shorter coiled-coils in CnuC, CnuD, and CnuH homologs were less strongly, but nevertheless confidently predicted, usually being detected by HHpred and by at least one but typically, more than one, coiled-coil prediction tool. The analysis was performed on both representative individual sequences and consensus sequences. The potential coiled-coils in CnuE homologs were often only found by Ncoils and were near the limit of detection, but these regions were also reported as coiled-coils in another study (*Anantharaman et al., 2012*). Given the context of extensive, high-probability coiled-coils in other components of the CoCoNuT systems with which they might interact, we chose to report these CnuE regions as coiled-coils, despite the comparative weakness of these predictions. In the AF2 multimer model of CnuHE, one of these potential coiled-coils is positioned near the coiled-coils detected in CnuH, suggesting they may facilitate interaction between these two factors.

## Preliminary phylogenetic analysis of CoCoNuT CnuH helicases

A comprehensive search and phylogenetic analysis of this family was beyond the scope of this work, but to determine the relationships between the CoCoNuTs and the rest of the UPF1-like helicases, we used the following procedure. We retrieved the best 2000 hits in each of two searches with UPF1 and CoCoNuT helicases as queries against both a database containing predicted proteins from 24,757 completely sequenced prokaryotic genomes downloaded from the NCBI GenBank in November 2021 and a database containing 72 representative eukaryotic genomes that were downloaded from the NCBI GenBank in June 2020. Next, we combined all proteins from the four searches, made a nonredundant set, and annotated them using CDD profiles, as described above. Then, we aligned them with MUSCLE v5 (*Edgar, 2022*), constructed an approximately maximum-likelihood tree with Fast-Tree, and mapped the annotations onto the tree. Genome neighborhoods were generated for these hits, as described above.

## Structural modeling with AlphaFold2 and searches for related structures

Protein structures were predicted using AlphaFold2 (AF2) v2.2.0 with local installations of complete databases required for AF2 (*Jumper et al., 2021*). Only single protein models with average predicted local distance difference test (pLDDT) scores ≥ 80 were retained for further analysis, and among the multimer models analyzed, all had average pLDDT scores ≥ 76, with only two scores <80 (*Mariani*

*et al., 2013*). Many of these models were used as queries to search for structurally similar proteins using DALI v5 against the Protein Data Bank (PDB) and using FoldSeek against the AlphaFold/UniProt50 v4, AlphaFold/Swiss-Prot v4, AlphaFold/Proteome v4, and PDB100 2201222 databases (*Holm, 2020*; *van Kempen et al., 2024*). Structure visualizations and comparisons were performed with ChimeraX (*Pettersen et al., 2021*) and the RCSB PDB website (*Berman et al., 2000*).

## Acknowledgements

The authors thank Becky Xu Hua Fu (University of California, San Francisco) for correspondence that led to her contribution of the name CoCoNuT, Andrew Z Fire and Usman Enam (Stanford University) for critical reading of the manuscript and insightful comments, Joseph Bondy-Denomy (University of California, San Francisco) for bringing the Shield factor ShdA to our attention, and Koonin group members for helpful discussions. The authors' research was supported by the Intramural Research Program of the National Institutes of Health (National Library of Medicine).

## Additional information

### Funding

| Funder | Grant reference number | Author |
|---|---|---|
| National Institutes of Health | Intramural Research Program | Eugene V Koonin |

The funders had no role in study design, data collection and interpretation, or the decision to submit the work for publication.

### Author contributions

Ryan T Bell, Conceptualization, Data curation, Investigation, Methodology, Writing – original draft, Writing – review and editing; Harutyun Sahakyan, Yuri I Wolf, Investigation, Methodology, Writing – review and editing; Kira S Makarova, Data curation, Investigation, Writing – review and editing; Eugene V Koonin, Conceptualization, Supervision, Investigation, Writing – original draft, Project administration, Writing – review and editing

### Author ORCIDs

Ryan T Bell http://orcid.org/0000-0003-1249-8398
Eugene V Koonin http://orcid.org/0000-0003-3943-8299

Reviewer #1 (Public review): https://doi.org/10.7554/eLife.94800.3.sa1
Reviewer #2 (Public review): https://doi.org/10.7554/eLife.94800.3.sa2
Author response https://doi.org/10.7554/eLife.94800.3.sa3

## Additional files

### Supplementary files

• Supplementary file 1. Protein structure prediction and analysis for CoCoNuT systems components.

• Supplementary file 2. List of AlphaFold 2 models for CoCoNuT protein components and their complexes, with modelarchive accession numbers.

• Supplementary file 3. GenBank accession numbers and protein sequences for protein components of the CoCoNuT systems.

• MDAR checklist

### Data availability

All data generated and analyzed in this study are included in the manuscript and supporting files. Domain identification statistics are listed in Supplementary file 1. The AlphaFold2 structural models generated and presented in the figures are available at https://modelarchive.org/ with accessions

listed in Supplementary file 2. The CoCoNuT systems are documented in detail in Supplementary file 3. The code generated during this work is available at https://doi.org/10.5281/zenodo.10971641.

The following dataset was generated:

| Author(s) | Year | Dataset title | Dataset URL | Database and Identifier |
|---|---|---|---|---|
| Bell et al. | 2024 | CoCoNuTs are a diverse subclass of Type IV restriction systems predicted to target RNA | https://doi.org/10.5281/zenodo.10971641 | Zenodo, 10.5281/zenodo.10971641 |

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
