## [Editor Report · eLife assessment]

This article marks a **fundamental** advance in our understanding of prokaryotic Type IV restriction systems. The authors provide an encyclopedic overview of a hitherto uncharacterized branch of these systems, which they name CoCoNuTs, for coiled-coil nuclease tandems. They provide **compelling** evidence that these nucleases target RNA and are part of an echeloned defense response following viral infection. This article will be of great interest to scientists studying prokaryotic immunity mechanisms, as well as broadly to protein scientists engaged in the analysis, classification, and functional annotation of the proteome of life.

---

## [Referee Report · Reviewer #1 (Public review)]

Summary:

In this manuscript, Bell et al. provide an exhaustive and clear description of the diversity of a new class of predicted type IV restriction systems that the authors denote as CoCoNuTs, for their characteristic presence of coiled-coil segments and nuclease tandems. Along with a comprehensive analysis that includes phylogenetics, protein structure prediction, extensive protein domain annotations, and in-depth investigation of encoding genomic contexts, they also provide detailed hypothesis about the biological activity and molecular functions of the members of this class of predicted systems. This work is highly relevant, it underscores the wide diversity of defence systems that are used by prokaryotes and demonstrates that there are still many systems to be discovered. The work is sound and backed up by a clear and reasonable bioinformatics approach.

Strengths:

The analysis provided by the authors is extensive and covers the three most important aspects that can be covered computationally when analysing a new family/superfamily: phylogenetics, genomic context analysis, and protein-structure-based domain content annotation. With this, one can directly have an idea about the superfamily of the predicted system and infer about their biological role. The bioinformatics approach is sound and makes use of the most current advances in the fields of protein evolution and structural bioinformatics.

Weaknesses:

It is not clear how coiled-coil segments were assigned if only based on AF2-predicted models or also backed by sequence analysis, as no description is provided in the methods. The structure prediction quality assessment is based solely on the average pLDDT of the obtained models (with a threshold of 80 or better). However, this is not enough, particularly when multimeric models were used. The PAE matrix should be used to evaluate relative orientations, particularly in the case where there is a prediction that parts from 2 proteins are interacting. In the case of multimers, interface quality scores, as the ipTM or pDockQ, should also be considered and, at minimum, reported.

These weaknesses were addressed during revision, and the results provided by the authors support their conclusions. The data resulting from this work will be useful for the general life sciences community, particularly the prokaryotic defense and microbiology communities. It also underscores the high range of functionally unknowns in sequenced genomes that are now much easier to find and interpret due to the success of deep-learning based methods and automated robust bioinformatics pipelines.

---

## [Referee Report · Reviewer #2 (Public review)]

Summary:

In this work, using in-depth computational analysis, Bell et al. explore the diverse repertoire of type IV McrBC modification dependent restriction systems. The prototypical two-component McrBC system has been structurally and functionally characterised and is known to act as a defence by restricting phage and foreign DNA containing methylated cytosines. Here, the authors find previously unanticipated complexity and versatility of these systems and focus on detailed analysis and classification of a distinct branch, the so-called CoCoNut, named after its composition of coiled-coil structures and tandem nucleases. These CoCoNut systems are predicted to target RNA as well as DNA and to utilise defence mechanisms with some similarity to type III CRISPR-Cas systems.

Strengths:

This work is enriched with a plethora of ideas and a myriad of compelling hypotheses that now will await experimental verification. The study comes from the group that was amongst the first to describe, characterise, and classify CRISPR-Cas systems. By analogy, the findings described here can similarly promote ingenious experimental and conceptual research that could further drive technological advances. It could also instigate vigorous scientific debates that will ultimately benefit the community.

Weaknesses:

The multi-component systems described here function in the context of large oligomeric complexes similarly to the prototypical McrBC system. While the AlphaFold2 (AF2) multimer predictions are provided in this work, these are not compared with the known McrBC structures. These comparisons could have been helpful not only for providing insights into these multimeric protein systems but also for giving more sound explanations of the differences observed amongst different McrBC types.

---

## [Author Response]

The following is the authors’ response to the original reviews.

**Public Reviews:**

**Reviewer #1 (Public Review):**
Summary:In this manuscript, Bell et al. provide an exhaustive and clear description of the diversity of a new class of predicted type IV restriction systems that the authors denote as CoCoNuTs, for their characteristic presence of coiled-coil segments and nuclease tandems. Along with a comprehensive analysis that includes phylogenetics, protein structure prediction, extensive protein domain annotations, and an in-depth investigation of encoding genomic contexts, they also provide detailed hypotheses about the biological activity and molecular functions of the members of this class of predicted systems. This work is highly relevant, it underscores the wide diversity of defence systems that are used by prokaryotes and demonstrates that there are still many systems to be discovered. The work is sound and backed-up by a clear and reasonable bioinformatics approach. I do not have any major issues with the manuscript, but only some minor comments.Strengths:The analysis provided by the authors is extensive and covers the three most important aspects that can be covered computationally when analysing a new family/superfamily: phylogenetics, genomic context analysis, and protein-structure-based domain content annotation. With this, one can directly have an idea about the superfamily of the predicted system and infer their biological role. The bioinformatics approach is sound and makes use of the most current advances in the fields of protein evolution and structural bioinformatics.Weaknesses:It is not clear how coiled-coil segments were assigned if only based on AF2-predicted models or also backed by sequence analysis, as no description is provided in the methods. The structure prediction quality assessment is based solely on the average pLDDT of the obtained models (with a threshold of 80 or better). However, this is not enough, particularly when multimeric models are used. The PAE matrix should be used to evaluate relative orientations, particularly in the case where there is a prediction that parts from 2 proteins are interacting. In the case of multimers, interface quality scores, such as the ipTM or pDockQ, should also be considered and, at minimum, reported.

A description of the coiled-coil predictions has been added to the Methods. For multimeric models, PAE matrices and ipTM+pTM scores have been included in Supplementary Data File S1.

**Reviewer #2 (Public Review):**
Summary:In this work, using in-depth computational analysis, Bell et al. explore the diverse repertoire of type IV McrBC modification-dependent restriction systems. The prototypical two-component McrBC system has been structurally and functionally characterised and is known to act as a defence by restricting phage and foreign DNA containing methylated cytosines. Here, the authors find previously unanticipated complexity and versatility of these systems and focus on detailed analysis and classification of a distinct branch, the so-called CoCoNut, named after its composition of coiled-coil structures and tandem nucleases. These CoCoNut systems are predicted to target RNA as well as DNA and to utilise defence mechanisms with some similarity to type III CRISPR-Cas systems.Strengths:This work is enriched with a plethora of ideas and a myriad of compelling hypotheses that now await experimental verification. The study comes from the group that was amongst the first to describe, characterize, and classify CRISPR-Cas systems. By analogy, the findings described here can similarly promote ingenious experimental and conceptual research that could further drive technological advances. It could also instigate vigorous scientific debates that will ultimately benefit the community.Weaknesses:The multi-component systems described here function in the context of large oligomeric complexes. Some of the single chain AF2 predictions shown in this work are not compatible, for example, with homohexameric complex formation due to incompatible orientation of domains. The recent advances in protein structure prediction, in particular AlphaFold2 (AF2) multimer, now allow us to confidently probe potential protein-protein interactions and protein complex formation. This predictive power could be exploited here to produce a better glimpse of these multimeric protein systems. It can also provide a more sound explanation for some of the observed differences amongst different McrBC types.

Hexameric CnuB complexes with CnuC stimulatory monomers for Type I-A, I-B, I-C, II, and III-A CoCoNuT systems have been modeled with AF2 and included in Supplementary Data File S1, albeit without the domains fused to the GTPase N-terminus (with the exception of Type I-B, which lacks the long coiled-coil domain fused to the GTPase and was modeled with its entire sequence). Attempts to model the other full-length CnuB hexamers did not lead to convincing results.

**Recommendations for the authors:**

**Reviewing Editor:**
The detailed recommendations by the two reviewers will help the authors to further strengthen the manuscript, but two points seem particularly worth considering: 1. The methods are barely sketched in the manuscript, but it could be useful to detail them more closely. Particularly regarding the coiled-coil segments, which are currently just statists, useful mainly for the name of the family, more detail on their prediction, structural properties, and purpose would be very helpful. 2. Due to its encyclopedic nature, the wealth of material presented in the paper makes it hard to penetrate in one go. Any effort to make it more accessible would be very welcome. Reviewer 1 in particular has made a number of suggestions regarding the figures, which would make them provide more support for the findings described in the text.

A description of the techniques used to identify coiled-coil segments has been added to the Methods. Our predictions ranged from near certainty in the coiled-coils detected in CnuB homologs, to shorter helices at the limit of detection in other factors. We chose to report all probable coiled-coils, as the extensive coiled-coils fused to CnuB, which are often the only domain present other than the GTPase, imply involvement in mediating complex formation by interacting with coiled-coils in other factors, particularly the other CoCoNuT factors. The suggestions made by Reviewer 1 were thoughtful and we made an effort to incorporate them.

**Reviewer #1 (Recommendations For The Authors):**
I do not have any major issues with the manuscript. I have however some minor comments, as described below.The last sentence of the abstract at first reads as a fact and not a hypothesis resulting from the work described in the manuscript. After the second read, I noticed the nuances in the sentence. I would suggest a rephrasing to emphasize that the activity described is a theoretical hypothesis not backed-up by experiments.

This sentence has been rephrased to make explicit the hypothetical nature of the statement.

In line 64, the authors rename DUF3578 as ADAM because indeed its function is not unknown. Did the authors consider reaching out to InterPro to add this designation to this DUF? A search in interpro with DUF3578 results in "MrcB-like, N-terminal domain" and if a name is suggested, it may be worthwhile to take it to the IntrePro team.

We will suggest this nomenclature to InterPro.

I find Figure 1E hard to analyse and think it occupies too much space for the information it provides. The color scheme, the large amount of small slices, and the lack of numbers make its information content very small. I would suggest moving this to the supplementary and making it instead a bar plot. If removed from Figure 1, more space is made available for the other panels, particularly the structural superpositions, which in my opinion are much more important.

We have removed Figure 1E from the paper as it adds little information beyond the abundance and phyletic distribution of sequenced prokaryotes, in which McrBC systems are plentiful.

In Figure 2, it is not clear due to the presence of many colorful "operon schemes" that the tree is for a single gene and not for the full operon segment. Highlighting the target gene in the operons or signalling it somehow would make the figure easy to understand even in the absence of the text and legend. The same applies to Supplementary Figure 1.

The legend has been modified to show more clearly that this is a tree of McrB-like GTPases.

In line 146, the authors write "AlphaFold-predicted endonucelase fold" to say that a protein contains a region that AF2 predicts to fold like an endonuclease. This is a weird way of writing it and can be confusing to non-expert readers. I would suggest rephrasing for increased clarity.

This sentence has been rephrased for greater clarity.

In line 167, there is a [47]. I believe this is probably due to a previous reference formatting.

Indeed, this was a reference formatting error and has been fixed.

In most figures, the color palette and the use of very similar color palettes for taxonomy pie charts, genomic context composition schemes, and domain composition diagrams make it really hard to have a good understanding of the image at first. Legends are often close to each other, and it is not obvious at first which belong to what. I would suggest changing the layouts and maybe some color schemes to make it easier to extract the information that these figures want to convey.

It seemed that Figure 4 was the most glaring example of these issues, and it has been rearranged for easier comprehension.

In the paragraph that starts at line 199, the authors mention an Ig-like domain that is often found at the N-terminus of Type I CoCoNuTs. Are they all related to each other? How conserved are these domains?

These domains are all predicted to adopt a similar beta-sandwich fold and are found at the N-terminus of most CoCoNuT CnuC homologs, suggesting they are part of the same family, but we did not undertake a more detailed sequenced-based analysis of these regions.

We also find comparable domains in the CnuC/McrC-like partners of the abundant McrB-like NxD motif GTPases that are not part of CoCoNuT systems, and given the similarity of some of their predicted structures to Rho GDP-dissociation inhibitor 1, we suspect that they have coevolved as regulators of the non-canonical NxD motif GTPase type. Our CnuBC multimer models showing consistent proximity between these domains in CnuC and CnuB GTPase domains suggest this could indeed be the case. We plan to explore these findings further in a forthcoming publication.

In line 210, the authors write "suggesting a role in overcrowding-induced stress response". Why so? In >all other cases, the authors justify their hypothesis, which I really appreciated, but not here.

A supplementary note justifying this hypothesis has been added to Supplementary Data File S1.

At the end of the paragraph that starts in line 264, the authors mention that they constructed AF2 multimeric models to predict if 2 proteins would interact. However, no quality scores were provided, particularly the PAE matrix. This would allow for a better judgement of this prediction, and I would suggest adding the PAE matrix as another panel in the figure where the 3D model of the complex is displayed.

The PAE matrix and ipTM+pTM scores for this and other multimer models have been added to Supplementary Data File S1. For this model in particular, the surface charge distribution of the model has been presented to support the role of the domains that have a higher PAE in RNA binding.

In line 306, "(supplementary data)" refers to what part of the file?

This file has been renamed Supplementary Table S3 and referenced as such.

In line 464, the authors suggest that ShdA could interact with CoCoNuTs. Why not model the complex as done for other cases? what would co-folding suggest?

As we were not able to convincingly model full-length CnuB hexamers with N-terminal coiled-coils, we did not attempt modeling of this hypothetical complex with another protein with a long coiled-coil, but it remains an interesting possibility.

In line 528, why and how were some genes additionally analyzed with HHPred?

Justification for this analysis has been added to the Methods, but briefly, these genes were additionally analyzed if there were no BLAST hits or to confirm the hits that were obtained.

In the first section of the methods, the first and second (particularly the second) paragraphs are extremely long. I would suggest breaking them to facilitate reading.

This change has been made.

In line 545, what do the authors mean by "the alignment (...) were analyzed with HHPred"?

A more detailed description of this step has been added to the Methods.

The authors provide the models they produced as well as extensive supplementary tables that make their data reusable, but they do not provide the code for the automated steps, as to excise target sequence sections out of multiple sequence alignments, for example.

The code used for these steps has been in use in our group at the NCBI for many years. It will be difficult to utilize outside of the NCBI software environment, but for full disclosure, we have included a zipped repository with the scripts and custom-code dependencies, although there are external dependencies as well such as FastTree and BLAST. In brief, it involves PSI-BLAST detection of regions with the most significant homology to one of a set of provided alignments (seals-2-master/bin/wrappers/cog_psicognitor). In this case, the reference alignments of McrB-like GTPases and DUF2357 were generated manually using HHpred to analyze alignments of clustered PSI-BLAST results. This step provided an output of coordinates defining domain footprints in each query sequence, which were then combined and/or extended using scripts based on manual analysis of many examples with HHpred (footprint_finders/get_GTPase_frags.py and footprint_finders/get_DUF2357_frags.py), then these coordinates were used to excise such regions from the query amino acid sequence with a final script (seals-2-master/bin/misc/fa2frag).

**Reviewer #2 (Recommendations For The Authors):**
(1) Page 4, line 77 - 'PUA superfamily domains' could be more appropriate to use instead of "EVE superfamily".

While this statement could perhaps be applied to PUA superfamily domains, our previous work we refer to, which strongly supports the assertion, was restricted to the EVE-like domains and we prefer to retain the original language.

(2) Page 5. lines 128-130 - AF2 multimer prediction model could provide a more sound explanation for these differences.

Our AF2 multimer predictions added in this revision indeed show that the NxD motif McrB-like CoCoNuT GTPases interact with their respective McrC-like partners such that an immunoglobulin-like beta-sandwich domain, fused to the N-termini of the McrC homologs and similar to Rho GDP-dissociation inhibitor 1, has the potential to physically interact with the GTPase variants. However, we did not probe this in greater detail, as it is beyond the scope of this already highly complex article, but we plan to study it in the future.

(3) Page 8, line 252 - The surface charge distribution of CnuH OB fold domain looks very different from SmpB (pdb3iyr). In fact, the regions that are in contact with RNA in SmpB are highly acidic in CoCoNut CnuH. Although it looks likely that this domain is involved in RNA binding, the mode of interaction should be very different.

We did not detect a strong similarity between the CnuH SmpB-like SPB domain and PDB 3IYR, but when we compare the surface charge distribution of PDB 1WJX and the SPB domain, while there is a significant area that is positively charged in 1WJX that is negatively charged in SPB, there is much that overlaps with the same charge in both domains.

The similarity between SmpB and the SPB domain is significant, but definitely not exact. An important question for future studies is: If the domains are indeed related due to an ancient fusion of SmpB to an ancestor of CnuH, would this degree of divergence be expected?

In other words, can we say anything about how the function of a stand-alone tmRNA-binding protein could evolve after being fused to a complex predicted RNA helicase with other predicted RNA binding domains already present? Experimental validation will ultimately be necessary to resolve these kinds of questions, but for now, it may be safe to say that the presence of this domain, especially in conjunction with the neighboring RelE-like RTL domain and UPF1-like helicase domain, signals a likely interaction with the A-site of the ribosome, and perhaps restriction of aberrant/viral mRNA.